# Communication-Efficient Distributionally Robust Decentralized Learning

**Matteo Zecchin**                                   *matteo.zecchin@eurecom.fr*
*Communication Systems Department*
*EURECOM, Sophia Antipolis, France*

**Marios Kountouris**                                *marios.kountouris@eurecom.fr*
*Communication Systems Department*
*EURECOM, Sophia Antipolis, France*

**David Gesbert**                                    *david.gesbert@eurecom.fr*
*Communication Systems Department*
*EURECOM, Sophia Antipolis, France*

**Reviewed on OpenReview:** *https://openreview.net/forum?id=tnRRHzZPMq*

## Abstract

Decentralized learning algorithms empower interconnected devices to share data and computational resources to collaboratively train a machine learning model without the aid of a central coordinator. In the case of heterogeneous data distributions at the network nodes, collaboration can yield predictors with unsatisfactory performance for a subset of the devices. For this reason, in this work, we consider the formulation of a distributionally robust decentralized learning task and we propose a decentralized single loop gradient descent/ascent algorithm (AD-GDA) to directly solve the underlying minimax optimization problem. We render our algorithm communication-efficient by employing a compressed consensus scheme and we provide convergence guarantees for smooth convex and non-convex loss functions. Finally, we corroborate the theoretical findings with empirical results that highlight AD-GDA's ability to provide unbiased predictors and to greatly improve communication efficiency compared to existing distributionally robust algorithms.

## 1 Introduction

Decentralized learning algorithms have gained an increasing level of attention, mainly due to their ability to harness, in a fault-tolerant and privacy-preserving manner, the large computational power and data availability at the network edge (Chen & Sayed, 2012; Yan et al., 2012). In this framework, a set of interconnected nodes (smartphones, IoT devices, health centers, research labs, etc.) collaboratively train a machine learning model alternating between local model updates, based on in situ data, and peer-to-peer type of communication to exchange model-related information. Compared to federated learning in which a swarm of devices communicates with a central parameter server at each communication round, fully decentralized learning has the benefits of removing the single point of failure and of alleviating the communication bottleneck inherent to the star topology.

The heterogeneity of distributedly generated data entails a major challenge, represented by the notions of fairness (Dwork et al., 2012) and robustness (Quiñonero-Candela et al., 2009). In the distributed setup, the customary global loss function is the weighted sum of the local empirical losses, with each term weighted by the fraction of samples that the associated node stores. However, in the case of data heterogeneity across participating parties, a model minimizing such definition of risk can lead to unsatisfactory and unfair [1]

---

[1] In the machine learning community, the notion of fairness has many facets. In this work, we will use the term "fair" in accordance with the notion of good-intent fairness as introduced in (Mohri et al., 2019).

inference capabilities for certain subpopulations. Consider, for example, a consortium of hospitals spread across the world sharing medical data to devise a new drug and in which a small fraction of hospitals have medical records influenced by geographical confounders, such as local diet, meteorological conditions, etc. In this setting, a model obtained by myopically minimizing the standard notion of risk defined over the aggregated data can be severely biased towards some populations at the expense of others. This can lead to a potentially dangerous or unfair medical treatment as shown in Figure 2.

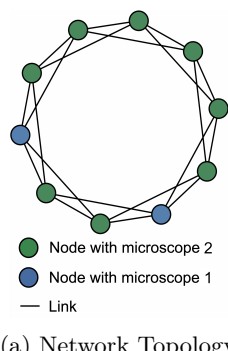

(a) Network Topology

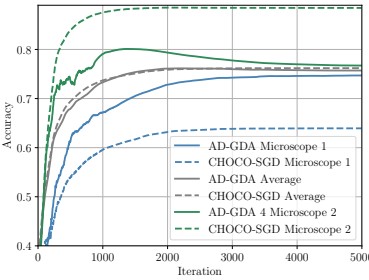

(b) Worst-node accuracy

Figure 2: Comparison between standard and distributionally robust decentralized learning procedures. We consider a network of 10 nodes that collaboratively train a mouse cell image classifier based on the Cells Out Of Sample 7-Class (COOS7) data set (Lu et al., 2019). Two nodes store samples obtained using a different microscope from the rest of the network devices (left figure). On the right, we report the validation accuracy of the standard decentralized learning (CHOCO-SGD) and the proposed *distributionally robust* decentralized learning (AD-GDA). We consider three different validation sets: one made of samples from one microscope, another with samples from the other, and one made of a 50/50 mixture of the two. CHOCO-SGD (dashed lines) yields a final model with a 24% accuracy gap between the two types of instruments. On the contrary, AD-GDA (solid lines) reduces the accuracy gap to less than 2% and improves fairness among collaborating parties. The average performance is not affected by the distributionally robust procedure.

To tackle this issue, distributionally robust learning aims at maximizing the worst-case performance over a set of distributions, termed an uncertainty set, which possibly contains the testing distribution of interest. Typical choices of the uncertainty sets are balls centered around the training distribution (Esfahani & Kuhn, 2018) or, whenever the training samples come from a mixture of distributions, the set of potential subpopulations resulting in such mixture (Duchi et al., 2019; Duchi & Namkoong, 2018). Robust distributed learning with heterogeneous data in which different distributions exist at the various devices falls in the latter category, as the natural ambiguity set is the one represented by the convex combination of the local distributions. In that case, minimizing the worst-case risk is equivalent to trying to ensure a minimum level of performance for each participating device. Specifically for the federated case, Mohri et al. (Mohri et al., 2019) introduced agnostic federated learning (AFL) as a means to ensure fairness and proposed a gradient-based algorithm to solve the distributionally robust optimization problem. In (Deng et al., 2021) a communication-efficient version of AFL, which avoids frequent retransmission of the dual variables, was proposed. More recently, distributionally robust learning has also been studied in the fully decentralized case. In this setting, the underlying minimax optimization problem becomes challenging and distributionally robust learning has been limited to a special formulation — Kullback-Leibler (KL) regularization — that allows simplifying the problem (Issaid et al., 2022).

In virtue of the advantages of the fully decentralized setup and advocating the necessity for robust and fair predictors, in this work we propose AD-GDA, a novel distributionally robust algorithm for the decentralized setting. In contrast to previous works, our solution *directly* tackles the minimax optimization problem in a *fully decentralized* fashion. Our algorithm is general and encompasses previous solutions as particular choices of the regularization function. Furthermore, we show that it is possible to perform distributionally robust optimization in a communication-efficient manner by employing a compressed gossip scheme without hampering the rate of convergence of the algorithm.

Table 1: Comparison between the proposed and existing distributionally robust algorithms.

| | Features | | | Convergence rate | |
|---|---|---|---|---|---|
| | Topologies | Compression | Regularizer | Convex | Non-convex |
| AD-GDA (ours) | **Connected** | ✓ | **Strongly-concave** | $\mathcal{O}(T^{-1/2})$ | $\mathcal{O}(T^{-1/2})$ |
| DRFA (Deng et al., 2021) | Star | ✗ | **Strongly-concave** | $\mathcal{O}(T^{-3/8})$ | $\mathcal{O}(T^{-1/8})$ |
| DR-DSGD (Issaid et al., 2022) | **Connected** | ✗ | Kullback-Leibler | ✗ | $\mathcal{O}(T^{-1/2})$ |

**Contributions:** The main contributions of the work are the following:

- We propose AD-GDA, an optimization algorithm to perform distributionally robust learning in a fully decentralized fashion. As detailed in Table 1, previous works have either been limited to the federated case or particular choices of the regularizer, AD-GDA *directly* tackles the distributionally robust minimax optimization problem in a *fully decentralized* fashion. Despite the additional complexity stemming from solving the decentralized minimax optimization problem, our solution is computation and communication efficient. AD-GDA alternates between local single-loop stochastic gradient descent/ascent model updates and compressed consensus steps to cope with local connectivity in a communication-efficient manner.

- We establish convergence guarantees for the proposed algorithm both in the case of smooth convex and smooth non-convex local loss functions that match or outperform the existing ones (see Table 1). In the former case, the algorithm returns an $\epsilon$-optimal solution after $\mathcal{O}(1/\epsilon^2)$ iterations. In the latter, the output is guaranteed to be an $\epsilon$-stationary solution after $\mathcal{O}(1/\epsilon^2)$ iterations whenever the stochastic gradient variance is also bounded by $\epsilon$, otherwise, we can obtain the same guarantee by increasing the number of calls to the stochastic gradient oracle.

- We empirically demonstrate AD-GDA capability in finding robust predictors under different compression schemes, network topologies, and models. First, we compare the proposed algorithm with compressed decentralized stochastic gradient descent (CHOCO-SGD) and highlight the merits of the distributionally robust procedure. We then consider the existing distributionally robust learning algorithms; namely, Distributionally Robust Federated Averaging (DRFA) (Deng et al., 2021) and Distributionally Robust Decentralized Stochastic Gradient Descent (DR-DSGD) (Issaid et al., 2022). We show that AD-GDA attains the same worst-case distribution accuracy transmitting a fraction of the bits and it is up to 4× and 10× times more communication efficient compared to DRFA and DR-DSGD, respectively.

## 2   Related Work

**Communication-efficient decentralized learning.**   Initiated in the 80s by the work of Tsitsiklis (Tsitsiklis, 1984; Tsitsiklis et al., 1986), the study of decentralized optimization algorithms was spurred by their adaptability to various network topologies, reliability to link failures, privacy-preserving capabilities, and potentially superior convergence properties compared to the centralized counterpart (Chen & Sayed, 2012; Yan et al., 2012; Olfati-Saber et al., 2007; Ling et al., 2012; Lian et al., 2017). This growing interest and the advent of large-scale machine learning brought forth an abundance of optimization algorithms both in the deterministic and stochastic settings (Nedic & Ozdaglar, 2009; Wei & Ozdaglar, 2012; Duchi et al., 2011; Shamir & Srebro, 2014; Rabbat, 2015). With the intent of extending its applicability, a concurrent effort has been made to devise techniques able to reduce the delay due to inter-node communication. Notable results in this direction are the introduction of message compression techniques, such as sparsification and quantization (Stich et al., 2018; Aji & Heafield, 2017; Alistarh et al., 2018; 2017; Bernstein et al., 2018; Koloskova et al., 2019b), and event-triggered communication to allow multiple local updates between communication rounds (Stich, 2018; Yu et al., 2019)

**Distributional robust learning.** Tracing back to the work of Scarf (Scarf, 1957), distributional robustness copes with the frequent mismatch between training and testing distributions by posing the training process as a game between a learner and an adversary, which has the ability to choose the testing distribution within an uncertainty set. Restraining the decisional power of the adversary is crucial to obtain meaningful and tractable problems and a large body of the literature deals with uncertainty sets, represented by balls centered around the training distribution and whose radii are determined by $f$-divergences (Namkoong & Duchi, 2016; Hu & Hong, 2013) or Wasserstein distance (Wozabal, 2012; Jiang & Guan, 2016; Esfahani & Kuhn, 2018). As such, distributional robustness is deeply linked to stochastic game literature. In this context, Lin et al. (2020b) provides last-iterate guarantees in case of coercive games. Refined results are later obtained in (Loizou et al., 2021), for expected coercive games and by relaxing the bounded gradient assumption. Furthermore, distributional robustness is deeply linked with the notion of fairness as particular choices of uncertainty sets allow guaranteeing uniform performance across the latent subpopulations in the data (Duchi & Namkoong, 2018; Duchi et al., 2019). In the case of federated learning, robust optimization ideas have been explored to ensure uniform performance across all participating devices (Mohri et al., 2019). Distributionally robust learning has also been studied in the fully decentralized scenario in the case of Kullback-Leibler (KL) regularizers for which exists an exact solution for the inner maximization problem (Issaid et al., 2022).

**Decentralized minimax optimization.** Saddle point optimization algorithms are of great interest given their wide range of applications in different fields of machine learning, including generative adversarial networks (Goodfellow et al., 2014), robust adversarial training (Sinha et al., 2017; Madry et al., 2017), and multi-agent reinforcement learning (Pinto et al., 2017; Li et al., 2019). Their convergence properties have also been studied in the decentralized scenario for the convex-concave setting (Koppel et al., 2015; Mateos-Núñez & Cortés, 2015). More recently, the assumptions on the convexity and concavity of the objective function have been relaxed. In Tsaknakis et al. (2020) an algorithm for nonconvex strongly-concave objective functions has been proposed; however, the double-loop nature of the solution requires solving the inner maximization problem with an increasing level of accuracy rendering it potentially slow. On the other hand, our algorithm is based on a single loop optimization scheme - with dual and primal variables being updated at each iteration in parallel - and, consequently, has a lower computational complexity. For the nonconvex-nonconcave case, Liu et al. (2019b) provides a proximal point algorithm while a simpler gradient-based algorithm is provided in Liu et al. (2019a) to train generative adversarial networks in a decentralized fashion. None of these works take into consideration communication efficiency in their algorithms.

## 3 Preliminaries

**Distributed system** We consider a network of $m$ devices in which each node $i$ is endowed with a local objective function $f_i : \mathbb{R}^d \to \mathbb{R}$ given by $\mathbb{E}_{z \sim P_i} \ell(\boldsymbol{\theta}, z)$, with $P_i$ denoting the local distribution at node $i$ and $\boldsymbol{\theta} \in \mathbb{R}^d$ being the model parameter to be optimized. Whenever $P_i$ is replaced by an empirical measure $\hat{P}_{i,n_i}$, the local objective function coincides with the empirical risk computed over $n_i$ samples. Network nodes are assumed to be interconnected by a communication topology specified by a connected graph $\mathcal{G} := (\mathcal{V}, \mathcal{E})$ in which $\mathcal{V} = \{1, \ldots, m\}$ indexes the devices and $(i, j) \in \mathcal{E}$ if and only if nodes $i$ and $j$ can communicate. For each node $i \in \mathcal{V}$, we define its set of neighbors by $\mathcal{N}(i) := \{j : (i, j) \in \mathcal{E}\}$ and since we assume self-communication we have $(i, i) \in \mathcal{N}(i)$ for all $i$ in $\mathcal{V}$. At each communication round, the network nodes exchange messages with their neighbors and average the received messages according to a mixing matrix $W \in \mathbb{R}^{m \times m}$.

**Assumption 3.1.** The mixing matrix $W \in \mathbb{R}^{m \times m}$ is symmetric and doubly-stochastic; we denote its spectral gap — the difference between the moduli of the two largest eigenvalues — by $\rho \in (0, 1]$ and define $\beta = \|I - W\|_2 \in [0, 2]$.

**Compressed communication** Being the communication phase the major bottleneck of decentralized training, we assume that nodes transmit only compressed messages instead of sharing uncompressed model updates. To this end, we define a, possibly randomized, compression operator $Q : \mathbb{R}^d \to \mathbb{R}^d$ that satisfies the following assumption.

**Assumption 3.2.** For any $\boldsymbol{x} \in \mathbb{R}^n$ and for some $\delta \in (0, 1]$,

$$\mathbb{E}_{\mathbb{Q}} \left[ \|Q(\boldsymbol{x}) - \boldsymbol{x}\|^2 \right] \leq (1 - \delta) \|\boldsymbol{x}\|^2. \tag{1}$$

The above definition is quite general as it entails both biased and unbiased compression operators. For instance, random quantization (Alistarh et al., 2017) falls into the former class and satisfies (1) with $\delta = \frac{1}{\tau}$. For a given vector $\boldsymbol{x} \in \mathbb{R}^d$ and quantization levels $2^b$, it yields a compressed message

$$\boldsymbol{x}_b = \frac{\text{sign}(\boldsymbol{x}) \, \|\boldsymbol{x}\|}{2^b \tau} \left\lfloor 2^b \frac{|\boldsymbol{x}|}{\|\boldsymbol{x}\|} + \xi \right\rfloor \tag{2}$$

with $\tau = 1 + \min \left\{ d/2^{2b}, \sqrt{d}/2^b \right\}$ and $\xi \sim \mathcal{U}[0,1]^{\otimes d}$. A notable representative of the biased category is the top-$K$ sparsification (Stich et al., 2018), which for a given vector $\boldsymbol{x} \in \mathbb{R}^d$ returns the $K$ largest magnitude components and satisfies (1) with $\delta = \frac{K}{d}$. Operators of that type have been previously considered in the context of decentralized learning and the effect of compressed communication in decentralized stochastic optimization has been previously investigated (Koloskova et al., 2019a;b; Stich et al., 2018). The resulting communication cost savings have been showcased in the context of decentralized training of deep neural networks (Koloskova et al., 2019a). However, to the best of our knowledge, there are no applications of compressed communication to distributional robust training in the decentralized setup.

---

**Algorithm 1:** Agnostic Decentralized GDA with Compressed Communication (AD-GDA)

**Input**  : Number of nodes $m$, number of iterations $T$, learning rates $\eta_\theta$ and $\eta_\lambda$, mixing matrix $W$, initial values $\boldsymbol{\theta}^0 \in \mathbb{R}^d$ and $\boldsymbol{\lambda}^0 \in \Delta^{m-1}$.

**Output:** $\boldsymbol{\theta}_o = \frac{1}{T} \sum_{t=0}^{T-1} \bar{\boldsymbol{\theta}}^t$, $\boldsymbol{\lambda}_o = \frac{1}{T} \sum_{t=0}^{T-1} \bar{\boldsymbol{\lambda}}^t$

initialize $\boldsymbol{\theta}_i^0 = \boldsymbol{\theta}^0$, $\boldsymbol{\lambda}_i^0 = \boldsymbol{\lambda}^0$ and $\boldsymbol{s}_i^0 = 0$ for $i = 1, \ldots, m$

**for** $t$ **in** $0, \ldots T-1$ **do**

    // In parallel at each node $i$

    $\boldsymbol{\theta}_i^{t+\frac{1}{2}} \leftarrow \boldsymbol{\theta}_i^t - \eta_\theta \nabla_{\boldsymbol{\theta}} g_i(\boldsymbol{\theta}_i^t, \boldsymbol{\lambda}_i^t, \xi_i^t)$               // Descent Step

    $\boldsymbol{\lambda}_i^{t+\frac{1}{2}} \leftarrow \mathcal{P}_\Lambda \left( \boldsymbol{\lambda}_i^t + \eta_\lambda \nabla_{\boldsymbol{\lambda}} g_i(\boldsymbol{\theta}_i^t, \boldsymbol{\lambda}_i^t, \xi_i^t) \right)$       // Projected Ascent Step

    $\boldsymbol{\theta}_i^{t+1} \leftarrow \boldsymbol{\theta}_i^{t+\frac{1}{2}} + \gamma \left( \boldsymbol{s}_i^t - \hat{\boldsymbol{\theta}}_i^t \right)$                 // Gossip

    $\boldsymbol{q}_i^t \leftarrow Q \left( \boldsymbol{\theta}_i^{t+1} - \hat{\boldsymbol{\theta}}_i^t \right)$                    // Compression

    send $(\boldsymbol{q}_i^t, \boldsymbol{\lambda}_i^{t+\frac{1}{2}})$ to $j \in \mathcal{N}(i)$ and receive $(\boldsymbol{q}_j^t, \boldsymbol{\lambda}_j^{t+\frac{1}{2}})$ from $j \in \mathcal{N}(i)$    // Msgs exchange

    $\hat{\boldsymbol{\theta}}_i^{t+1} \leftarrow \boldsymbol{q}_i^t + \hat{\boldsymbol{\theta}}_i^t$               // Public variables update

    $\boldsymbol{s}_i^{t+1} \leftarrow \boldsymbol{s}_i^t + \sum_{j=1}^m w_{i,j} \boldsymbol{q}_j$

    $\boldsymbol{\lambda}_i^{t+1} \leftarrow \sum_{j=1}^m w_{i,j} \boldsymbol{\lambda}_j^{t+\frac{1}{2}}$             // Dual variable averaging

**end**

---

**Distributionally robust network loss**  In order to obtain a final predictor with satisfactory performance for all local distributions $\{P_i\}_{i=1}^m$, the common objective is to learn global model which is distributionally robust with respect to the ambiguity set $\mathcal{P} := \left\{ \sum_{i=1}^m \lambda_i P_i : \lambda_i \in \Delta^{m-1} \right\}$ where $\Delta^{m-1}$ where denotes the $m-1$ probability simplex. As shown in Mohri et al. (2019), a network objective function that effectively works as proxy for this scope is given by

$$\min_{\boldsymbol{\theta} \in \mathbb{R}^d} \max_{\boldsymbol{\lambda} \in \Delta^{m-1}} \left( g(\boldsymbol{\theta}, \boldsymbol{\lambda}) := \frac{1}{m} \sum_{i=1}^m \underbrace{(\lambda_i f_i(\boldsymbol{\theta}) + \alpha r(\boldsymbol{\lambda}))}_{:= g_i(\boldsymbol{\theta}, \boldsymbol{\lambda})} \right) \tag{3}$$

in which $r : \Delta^{m-1} \to \mathbb{R}$ is a strongly-concave regularizer and $\alpha \in \mathbb{R}^+$. For instance, in the empirical risk minimization framework in which each node $i$ is endowed with a training set $\mathcal{D}_i \sim P_i^{\otimes n_i}$ and the overall number of training points is $n = \sum_i n_i$, a common choice of $r(\boldsymbol{\lambda})$ is $\chi^2(\boldsymbol{\lambda}) := \sum_i \frac{(\lambda_i - n_i/n)^2}{n_i/n}$ or the Kullback-Leibler divergence $D_{\text{KL}}(\boldsymbol{\lambda}) := \sum_i \lambda_i \log(\lambda_i n/n_i)$ (Issaid et al., 2022). Restricting (3) to the latter regularizer, the inner maximization problem can be solved exactly (Issaid et al., 2022; Mohajerin Esfahani & Kuhn, 2018).

In what follows, we refer to $\boldsymbol{\theta}$ and $\boldsymbol{\lambda}$ as the primal and dual variables, respectively, and make the following fairly standard assumptions on the local functions $g_i$ and the stochastic oracles available at the network nodes.

**Assumption 3.3.** Each function $g_i(\boldsymbol{\theta}, \boldsymbol{\lambda})$ is differentiable in $\mathbb{R}^d \times \Delta^{m-1}$, $L$-smooth and $\mu$- concave in $\boldsymbol{\lambda}$.

**Assumption 3.4.** Each node $i$ has access to the stochastic gradient oracles $\nabla_{\boldsymbol{\theta}} g_i(\boldsymbol{\theta}, \boldsymbol{\lambda}, \xi_i)$ and $\nabla_{\boldsymbol{\lambda}} g_i(\boldsymbol{\theta}, \boldsymbol{\lambda}, \xi_i)$, with randomness w.r.t. $\xi_i$, which satisfy the following assumptions:

- Unbiasedness

$$\mathbb{E}_{\xi_i}\left[\nabla_{\boldsymbol{\theta}} g_i(\boldsymbol{\theta}, \boldsymbol{\lambda}, \xi_i)\right] = \nabla_{\boldsymbol{\theta}} g_i(\boldsymbol{\theta}, \boldsymbol{\lambda}), \quad \mathbb{E}_{\xi_i}\left[\nabla_{\boldsymbol{\lambda}} g_i(\boldsymbol{\theta}, \boldsymbol{\lambda}, \xi_i)\right] = \nabla_{\boldsymbol{\lambda}} g_i(\boldsymbol{\theta}, \boldsymbol{\lambda}). \tag{4}$$

- Bounded variance

$$\mathbb{E}_{\xi_i}\left[\left\|\nabla_{\boldsymbol{\theta}} g_i(\boldsymbol{\theta}, \boldsymbol{\lambda}, \xi_i) - \nabla_{\boldsymbol{\theta}} g_i(\boldsymbol{\theta}, \boldsymbol{\lambda})\right\|^2\right] \leq \sigma_{\theta}^2, \quad \mathbb{E}_{\xi_i}\left[\left\|\nabla_{\boldsymbol{\lambda}} g_i(\boldsymbol{\theta}, \boldsymbol{\lambda}, \xi_i) - \nabla_{\boldsymbol{\lambda}} g_i(\boldsymbol{\theta}, \boldsymbol{\lambda})\right\|^2\right] \leq \sigma_{\lambda}^2. \tag{5}$$

- Bounded magnitude

$$\mathbb{E}_{\xi_i}\left[\left\|\nabla_{\boldsymbol{\theta}} g_i(\boldsymbol{\theta}, \boldsymbol{\lambda}, \xi_i)\right\|^2\right] \leq G_{\theta}^2, \quad \mathbb{E}_{\xi_i}\left[\left\|\nabla_{\boldsymbol{\lambda}} g_i(\boldsymbol{\theta}, \boldsymbol{\lambda}, \xi_i)\right\|^2\right] \leq G_{\lambda}^2. \tag{6}$$

The above assumption implies that each network node can query stochastic gradients that are unbiased, have finite variance, and have bounded second moment. The bounded magnitude assumption is rather strong and limits the choice of regularization functions, but it is often made in distributed stochastic optimization (Stich et al., 2018; Koloskova et al., 2019a; Deng et al., 2021).

## 4 Distributionally Robust Decentralized Learning Algorithm

Problem (3) entails solving a distributed minimax optimization problem in which, at every round, collaborating nodes store a private value of the model parameters and the dual variable, which are potentially different from node to node. We denote the estimate of the primal and dual variables of node $i$ at time $t$ by $\boldsymbol{\theta}_i^t$ and $\boldsymbol{\lambda}_i^t$ and the network estimates at time $t$ as $\bar{\boldsymbol{\theta}}^t = \frac{1}{m} \sum_{i=1}^m \boldsymbol{\theta}_i^t$ and $\bar{\boldsymbol{\lambda}}^t = \frac{1}{m} \sum_{i=1}^m \boldsymbol{\lambda}_i^t$, respectively. The main challenge resulting from the decentralized implementation of the stochastic gradient descent/ascent algorithm consists in approaching a minimax solution or a stationary point (depending on the convexity assumption on the loss function) while concurrently ensuring convergence to a common global solution. To this end, the proposed procedure, given in Algorithm 1, alternates between a local update step and a consensus step. At each round, every node $i$ queries the local stochastic gradient oracle and, in parallel, updates the model parameter $\theta_i$ by a gradient descent step with learning rate $\eta_\theta > 0$ and the dual variable $\boldsymbol{\lambda}_i$ by a projected gradient ascent one with learning rate $\eta_\lambda > 0$ (Euclidean projection). Subsequently, a gossip strategy is used to share and average information between neighbors. In order to alleviate the communication burden of transmitting the vector of model parameters, which is typically high dimensional and contributes to the largest share of the communication load, a compressed gossip step is employed. To implement the compressed communication, we consider the memory-efficient version of CHOCO-GOSSIP (Koloskova et al., 2019b) in which each node needs to store only two additional variables $\hat{\boldsymbol{\theta}}_i$ and $\boldsymbol{s}_i$, each of the same size as $\boldsymbol{\theta}_i$. The first one is a public version of $\boldsymbol{\theta}_i$, while the second is used to track the evolution of the weighted average, according to matrix W, of the public variables at the neighboring nodes. Instead of transmitting $\boldsymbol{\theta}_i$, each node first computes an averaging step to update the value of the private value using the information about the public variables encoded in $\hat{\boldsymbol{\theta}}_i$ and $\boldsymbol{s}_i$. It then computes $\boldsymbol{q}_i$, a compressed representation of the difference between $\hat{\boldsymbol{\theta}}_i$ and $\boldsymbol{\theta}_i$, and shares it with the neighboring nodes to update the value of $\hat{\boldsymbol{\theta}}_i$ and $\boldsymbol{s}_i$ used in the averaging step in the next round. As the number of participating nodes is usually much smaller than the size of the model ($m \ll d$), the dual variable $\boldsymbol{\lambda}_i$ is updated sending uncompressed messages and then averaged according to matrix $W$. Note that AD-GDA implicitly assumes that collaborating parties are honest and for this reason, it does not employ any countermeasure against malicious nodes providing false dual variable information in order to steer the distributional robust network objective at their whim.

### 4.1 Comparison with existing algorithms

Before deriving the convergence guarantees for AD-GDA we compare the features of AD-GDA against other distributionally robust algorithms, DRFA and DR-DSGD (see Table 1). Firstly, we notice that AD-GDA and

DR-DSGD are fully decentralized algorithms, and therefore the only requirement for deployment is that the network is connected. In contrast, DRFA is a client-server algorithm and star topologies, which are known to be less fault tolerant and characterized by a communication bottleneck. Furthermore, AD-GDA is the only algorithm that attains communication efficiency by the means of message compression that, as we show in Section 5.2.2, allows AD-GDA to attain the same worst-node accuracy at a much lower communication cost compared to DRFA and DR-DSGD. It is also important to notice that DR-SGD is obtained by restricting the dual regularizer to Kullback-Leibler divergences, this allows sidestepping the minimax problem. On the other hand, AD-GDA directly tackles the distributionally robust problem (3) and therefore can be applied to any strongly concave regularizer that satisfies Assumption 3.4. For example, the chi-squared regularizer, which allows AD-GDA to direclty minimize the distributionally objective of Mohri et al. (2019).. In Section (4.3) we show AD-GDA recovers the same convergence rate of DR-DSGD in this more general set-up.

## 4.2 Convex Loss Function

We provide now a convergence guarantee for the solution output by Algorithm 1 for the case the loss function $\ell(\cdot)$ is convex in the model parameter $\boldsymbol{\theta}$. The result is given in the form of a sub-optimality gap bound for the function

$$\Phi(\boldsymbol{\theta}) = g\left(\boldsymbol{\theta}, \boldsymbol{\lambda}^*(\boldsymbol{\theta})\right), \quad \boldsymbol{\lambda}^*(\cdot) := \arg\max_{\boldsymbol{\lambda} \in \Delta^{m-1}} g(\cdot, \boldsymbol{\lambda}) \tag{7}$$

and it can be promptly derived from a primal-dual gap type of bound provided in the Appendix. In the bound we also refer to $\boldsymbol{\theta}^*(\cdot) \in \arg\max_{\boldsymbol{\theta} \in \mathbb{R}^d} g(\boldsymbol{\theta}, \cdot)$.

**Theorem 4.1.** *Under Assumptions 3.3, 3.4, we have that for any $\boldsymbol{\theta}^* \in \arg\min_{\boldsymbol{\theta}} \Phi(\boldsymbol{\theta})$ the solution $\boldsymbol{\theta}_o$ returned by Algorithm 1 with learning rates $\eta_\theta = \eta_\lambda = \frac{1}{\sqrt{T}}$ and consensus step size $\gamma = \frac{\rho^2 \delta}{16\rho + \rho^2 + 4\beta^2 + 2\rho\beta^2 - 8\rho\delta}$ satisfies*

$$\mathbb{E}\left[\Phi(\boldsymbol{\theta}_o) - \Phi(\boldsymbol{\theta}^*)\right] \leq \mathcal{O}\left(\frac{D_\theta + D_\lambda + G_\theta^2 + G_\lambda^2}{\sqrt{T}}\right) + \mathcal{O}\left(\frac{L D_\lambda G_\theta}{c\sqrt{T}} + \frac{L D_\theta G_\lambda}{\rho\sqrt{T}}\right) + \mathcal{O}\left(\frac{L G_\lambda^2}{\rho^2 T} + \frac{L G_\theta^2}{c^2 T}\right) \tag{8}$$

*where $D_\lambda := \max_t \mathbb{E}\left\|\bar{\boldsymbol{\lambda}}^t - \boldsymbol{\lambda}^*(\boldsymbol{\theta}_o)\right\|$, $D_\theta := \max_t \mathbb{E}\left\|\bar{\boldsymbol{\theta}}^t - \boldsymbol{\theta}^*(\boldsymbol{\lambda}_o)\right\|$ and $c = \frac{\rho^2 \delta}{82}$.*

The bound establishes a $\mathcal{O}(1/\sqrt{T})$ non-asymptotic optimality gap guarantee for the output solution. Compared to decentralized stochastic gradient descent (SGD) in the convex scenario, we obtain the same rate (without a dependency on the number of workers $m$) but with a dependency on the network topology and compression also in the lower order terms. Moreover, whenever $\boldsymbol{\theta}$ and $\boldsymbol{\lambda}$ are constrained in convex sets, the diameter of the two can be used to explicitly bound $D_\theta$ and $D_\lambda$.

## 4.3 Non-convex Loss Function

We now focus on the case where the relation between the model parameters $\boldsymbol{\theta}$ and the value of the loss function is non-convex. In this setting, carefully tuning the relation between primal and dual learning rates is key to establishing a convergent recursion. From the *centralized* two-time scale stochastic gradient descent literature, it is known that the primal learning rate $\eta_\theta$ has to be $1/(16(\kappa + 1))$ time smaller than the dual learning rate $\eta_\lambda$ (Lin et al., 2020a). The above relationship ensures that the objective function changes slowly enough in the dual variable $\boldsymbol{\lambda}$, and it allows to bound the distance between optimal values of $\boldsymbol{\lambda}^*(\boldsymbol{\theta}^t)$ and the current estimate $\boldsymbol{\lambda}^t$. However, in the *decentralized* setup, the estimates of $\boldsymbol{\theta}$ and $\boldsymbol{\lambda}$ differ at every node and therefore there exists multiple optimal values of the dual variable; namely, $\boldsymbol{\lambda}^*(\boldsymbol{\theta}_i^t)$ for $i = 1, \ldots, m$. Nonetheless, we find that it is sufficient to control the quantity $\delta_\lambda^t := \left\|\boldsymbol{\lambda}^*(\bar{\boldsymbol{\theta}}^t) - \bar{\boldsymbol{\lambda}}^t\right\|^2$; the distance between $\boldsymbol{\lambda}^*(\bar{\boldsymbol{\theta}}^t)$, the optimal dual variable for the *averaged* primal estimate, and $\bar{\boldsymbol{\lambda}}^t$, the current *averaged* dual estimate. The following lemma, whose proof can be found in Appendix A.3, allows us to characterize the behaviour of $\delta_\lambda^t$.

**Lemma 4.2.** *For $\eta_\theta = \frac{\eta_\lambda}{16(\kappa+1)^2}$ and $\eta_\lambda = \frac{1}{2L\sqrt{T}}$, the sequence of $\{\delta_\lambda^t\}_{t=1}^T$ generated by Algorithm 1 satisfies*

$$\sum_{t=1}^T \mathbb{E}\left[\delta_\lambda^t\right] \leq \frac{5\delta_\lambda^0 \kappa}{\eta_\lambda \mu} + \sum_{t=1}^T 5\kappa \left(4 D_\lambda^{t-1} \sqrt{\frac{1}{m}\mathbb{E}\left[\Xi_\theta^{t-1}\right]} + \frac{3\mathbb{E}\left[\Xi_\theta^{t-1}\right]}{m} + \frac{7\mathbb{E}\left[\Xi_\lambda^{t-1}\right]}{m}\right)$$

$$+ \sum_{t=1}^{T} 5 \left( \frac{8\kappa^2 \eta_\theta^2}{\eta_\lambda^2 \mu^2} \mathbb{E}\left[ \left\| \nabla \Phi(\bar{\theta}^{t-1}) \right\|^2 \right] \right) + 5T \left( \frac{2\eta_\lambda \sigma_\lambda^2}{m\mu} + \frac{4\sigma_\theta^2}{16^2 m(\kappa+1)^2 \mu^2} \right) \tag{9}$$

*where $\Xi_\theta$ and $\Xi_\lambda$ are the primal and dual consensus errors, and $D_\lambda^{t-1} = \left\| \bar{\lambda}^{t-1} - \lambda \right\|$.*

Lemma 4.2 provides us with an inequality that controls the "speed" at which the optimization problem changes from a network-level perspective. As such, the expression (9) contains consensus error terms that do not appear in the centralized setup. We find that to establish a convergent recursion while at the same time controlling the consensus error terms, the primal learning rate $\eta_\theta$ has to be $1/(16(\kappa+1)^2)$ time smaller than the dual $\eta_\lambda$ (therefore $1/(\kappa+1)$ smaller compared to the centralized case). Once this condition is met it is possible to provide a bound on the stationarity of the randomized solution, picked uniformly over time, that matches the one known for the *centralized* case.

**Theorem 4.3.** *Under Assumptions 3.3, 3.4, the iterates of Algorithm 1 with learning rates $\eta_\theta = \frac{\eta_\lambda}{16(\kappa+1)^2}$ and $\eta_\lambda = \frac{1}{2L\sqrt{T}}$ and consensus step size $\gamma = \frac{\rho^2 \delta}{16\rho + \rho^2 + 4\beta^2 + 2\rho\beta^2 - 8\rho\delta}$ satisfy*

$$\frac{\sum_{t=1}^{T} \mathbb{E}\left[ \left\| \nabla \Phi(\bar{\theta}^{t-1}) \right\|^2 \right]}{T} \le \mathcal{O}\left( L \frac{\Delta\Phi^T}{\sqrt{T}} + \frac{L^2 \kappa^2 D_\lambda^0}{2\sqrt{T}} \right) + \mathcal{O}\left( \frac{D_\lambda L G_\theta}{c\sqrt{T}} + \frac{\sigma_\theta^2 + \kappa\sigma_\lambda^2}{m\sqrt{T}} \right) + \mathcal{O}\left( \frac{G_\theta^2}{c^2 T} + \frac{\kappa G_\lambda^2}{\rho^2 T} \right) + \frac{\sigma_\theta^2}{m} \tag{10}$$

*where $\Delta\Phi^T = \mathbb{E}[\Phi(\bar{\theta}^0)] - \mathbb{E}[\Phi(\bar{\theta}^T)]$ and $c = \frac{\rho^2 \delta}{82}$.*

We note that the bound decreases at a rate $\mathcal{O}(1/\sqrt{T})$, except the last variance term which is non-vanishing. Nonetheless, whenever the variance of the stochastic gradient oracle for the primal variable is small or the number of participating devices is large, this term becomes negligible. Otherwise, at a cost of increased gradient complexity, each device can query the oracle $\mathcal{O}(1/\epsilon^2)$ times every round, average the results and make the stochastic gradient variance $\mathcal{O}(1/\epsilon^2)$. This procedure makes the bound vanish and leads to a gradient complexity matching the one of Sharma et al. (2022) given for the federated learning scenario.

Table 2: Final worst-case distribution accuracy attained by AD-GDA and CHOCO-SGD under different compression schemes and compression ratios.

| | Quantization | | | Sparsification | | |
|---|---|---|---|---|---|---|
| | 16 bit | 8 bit | 4 bit | 50% | 25% | 10% |
| Logistic AD-GDA | $59.19 \pm 2.05$ | $57.43 \pm 1.44$ | $55.75 \pm 2.09$ | $57.05 \pm 0.68$ | $54.02 \pm 1.14$ | $51.51 \pm 2.88$ |
| Logistic CHOCO-SGD | $30.69 \pm 0.96$ | $30.06 \pm 0.83$ | $29.46 \pm 0.05$ | $30.28 \pm 0.60$ | $28.56 \pm 0.54$ | $26.39 \pm 0.67$ |
| F.C. AD-GDA | $54.99 \pm 1.92$ | $48.99 \pm 2.30$ | $47.08 \pm 2.53$ | $51.85 \pm 2.11$ | $43.65 \pm 2.97$ | $38.95 \pm 3.21$ |
| F.C. CHOCO-SGD | $30.83 \pm 2.22$ | $28.08 \pm 2.50$ | $28.01 \pm 2.59$ | $29.92 \pm 2.54$ | $27.11 \pm 2.96$ | $25.91 \pm 3.20$ |

## 5 Experiments

In this section, we empirically evaluate AD-GDA capabilities in producing robust predictors. We first compare AD-GDA with CHOCO-SGD and showcase the merits of the distributionally robust procedure across different learning models, communication network topologies, and message compression schemes. We then consider larger-scale experimental setups in which we study the effect of the regularization on the worst-case distribution accuracy and compare AD-GDA against Distributionally Robust Federated Averaging (DRFA) (Deng et al., 2021) and Distributionally Robust Decentralized Stochastic Gradient Descent (DR-DSGD) (Issaid et al., 2022).

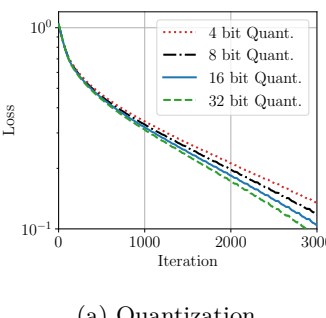
(a) Quantization

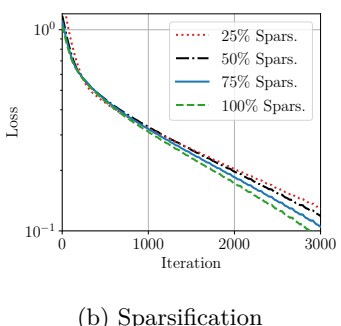
(b) Sparsification

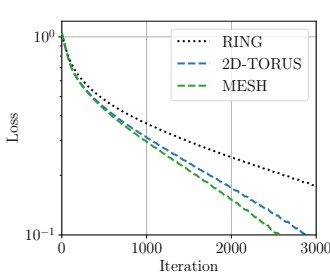

Figure 3: Convergence plots of AD-GDA for different quantization levels and schemes.

Figure 4: Convergence plot of AD-GDA for different topologies.

### 5.1 Fashion-MNIST Classification

We perform our experiments using the Fashion-MNIST data set (Xiao et al., 2017) [2], a popular data set made of images of 10 different clothing items, which is commonly used to test distributionally robust learners (Mohri et al., 2019; Deng et al., 2021). To introduce data heterogeneity, samples are partitioned across the network devices using a class-wise split. Namely, using a workstation equipped with a GTX 1080 Ti, we simulate a network of 10 nodes, each storing data points coming from one of the 10 classes. In this setting, we train a logistic regression model and a two-layer fully connected neural network with 25 hidden units to investigate both the convex and the non-convex cases. In both cases, we use the SGD optimizer and, to ensure consensus at the end of the optimization process, we consider a geometrically decreasing learning rate $\eta_\theta^t = r^{-t}\eta_\theta^0$ with ratio $r = 0.995$ and initial value $\eta_\theta^0 = 1$. The metrics that we track are the final worst-node distribution accuracy and the average accuracy over the aggregated data samples on the network estimate $\bar{\theta}^t$.

#### 5.1.1 Effect of Compression

We assess the effect of compression with a fixed budget in terms of communication rounds by organizing nodes in a ring topology and training the logistic model and the fully connected network for $T = 2000$ iterations. As a representative of the unbiased compression operators, we consider the $b$-bit random quantization scheme for $b = \{16, 8, 4\}$ bit levels, while for the biased category we implement the top-$K$ sparsification scheme saving $K = \{50\%, 25\%, 10\%\}$ of the original message components. For each compression scheme and compression level, we tune the consensus step size $\gamma$ by performing a grid search. We train the different models for 20 different random placements of the data shards across the devices using the distributionally robust and standard learning paradigms. In Table 2 we report the average worst-case accuracy attained by the final averaged model $\bar{\theta}^T$. AD-GDA almost doubles the worst-case accuracy compared to the not-robust baseline CHOCO-SGD (Koloskova et al., 2019b). This gain holds for both compression schemes and across different compression levels. For increased compression ratios, the worst-case accuracy degrades; however, for a comparable saving in communication bandwidth the unbiased quantization scheme results in superior performance than the biased sparsification compression operator. For a fixed optimization horizon compression degrades performance. Nonetheless, compression allows us to obtain the same accuracy level with fewer transmitted bits. In Figure 3 we report the convergence plot of AD-GDA under the different compression schemes and ratios. For this experiment we track the worst-node loss of a logistic model trained using a fixed learning rate $\eta_\theta = 0.1$. The convergence plots confirm the sub-linear rate of AD-GDA and highlight the effect of the compression level on the slope of convergence.

#### 5.1.2 Effect of Topology

We now turn to investigate the effect of node connectivity. Sparser communication topologies slow down the consensus process and therefore hamper the convergence of the algorithm. In the previous batch of

---

[2]The Fashion-MNIST data set is released under the MIT License

Table 3: Worst-node accuracy attained by AD-GDA and CHOCO-SGD for different network topologies.

| | Top-10% Sparsification | | 4-bit Quantization | |
|---|---|---|---|---|
| | 2D Torus | Mesh | 2D Torus | Mesh |
| Log. AD-GDA | $\mathbf{54.00 \pm 0.61}$ | $\mathbf{54.07 \pm 0.03}$ | $\mathbf{56.94 \pm 0.38}$ | $\mathbf{57.11 \pm 0.03}$ |
| Log. CHOCO-SGD | $26.82 \pm 0.41$ | $29.00 \pm 0.02$ | $30.82 \pm 0.24$ | $30.97 \pm 0.03$ |
| F.C. AD-GDA | $44.31 \pm 2.47$ | $45.21 \pm 2.22$ | $50.16 \pm 1.85$ | $50.80 \pm 1.83$ |
| F.C. CHOCO-SGD | $26.02 \pm 2.29$ | $26.38 \pm 2.65$ | $28.79 \pm 2.22$ | $28.96 \pm 1.87$ |

experiments, we considered a sparse ring topology, in which each node is connected to only two other nodes. Here, we explore two other network configurations with a more favorable spectral gap. The communication topology with each node connected to the other 4 nodes and the mesh case, in which all nodes communicate with each other. For these configurations, we consider the 4-bit quantization and top-10% sparsification compression schemes. In Table 3 we report the final worst-case performance for the different network configurations. As expected, network configurations with larger node degrees lead to higher worst-case accuracy owing to the faster convergence. In Figure 4 we provide the convergence plot of AD-GDA for different communication topologies. In particular, we track the worst-node loss versus the number of iterations for the logistic model optimized using a fixed learning rate $\eta_\theta = 0.1$. The predicted sublinear rate of the AD-GDA is confirmed and it is also possible to appreciate the influence of the spectral gap on the convergence rates.

## 5.2 Larger Scale Experiments

We now study AD-GDA and compared it with existing distributionally robust algorithms. We consider three larger-scale experimental setups:

- A larger scale version of the Fashion MNIST classification task of Section 5.1. In this section, we assume a larger network comprising a total of 50 devices with nodes storing samples from only one class.

- A CIFAR-10 image classification task based on 4-layer convolutional neural networks (CNN) (Krizhevsky et al., 2009). We consider a network of 20 network nodes and we introduce data heterogeneity by evenly partitioning the training set and changing the contrast of the images stored on the devices. The pixel value contrast $P \in [0, 255]$ is modified using the following non-linear transformation

$$f_c(P) = \text{clip}_{[0,255]}[(128 + c(P - 128))^{1.1}], \tag{11}$$

where $\text{clip}_{[0,255]}(\cdot)$ rounds values to the discrete set $[0, 255]$. For $c < 1$ the contrast is reduced while for $c > 1$ it is enhanced. We consider two network nodes storing images with reduced contrast ($c = 0.5$), two storing images with increased contrast ($c = 1.5$), and the rest of the nodes storing images with the original contrast level ($c = 1$). This setup can be used to model a camera network (e.g. surveillance network) containing devices deployed under different lighting conditions.

- A network of 10 nodes collaborating to train a 4-layer CNN to classify microscopy images from the biological data set COOS7 (Lu et al., 2019). Data heterogeneity is a consequence of the existence of different instruments used to sample the training data. In particular, we consider two of the collaborating nodes using a different microscope from the rest of the collaborating devices. This setup illustrates the risk of training models based on biological data affected by local confounders — in this case, the usage of different sensors.

### 5.2.1 Effect of Regularization

We first evaluate the effect of the regularization parameter $\alpha$ in the distributionally robust formulation (3) and study how it affects AD-GDA final performance. According to the two-player game interpretation of

Table 4: Testing accuracy attained by AD-DGA for different regularization values $\alpha$. For the Fashion-MNIST data set we report the worst and best class accuracy, for the CIFAR-10 data set the accuracy for the different contrast images and for the COOS7 data set the accuracy for the different microscopes. The last column in all tables is the the accuracy attained on a test data set comprising samples from all local data distributions.

(a) Fashion-MNIST.

|  | Worst Class | Best Class | Average |
|---|---|---|---|
| $\alpha = 10$ | $52.83 \pm 1.14$ | $\mathbf{90.63 \pm 0.18}$ | $\mathbf{77.31 \pm 0.19}$ |
| $\alpha = 1$ | $\mathbf{59.17 \pm 1.06}$ | $89.77 \pm 0.60$ | $76.77 \pm 0.04$ |
| $\alpha = 0.01$ | $58.47 \pm 0.92$ | $89.56 \pm 0.52$ | $76.67 \pm 0.09$ |

(b) COOS7.

|  | Microscope 1 | Microscope 2 | Average |
|---|---|---|---|
| $\alpha = 10$ | $66.93 \pm 0.23$ | $86.54 \pm 0.06$ | $76.73 \pm 0.13$ |
| $\alpha = 1$ | $72.27 \pm 0.31$ | $\mathbf{81.30 \pm 0.18}$ | $\mathbf{76.77 \pm 0.22}$ |
| $\alpha = 0.01$ | $\mathbf{75.00 \pm 0.24}$ | $75.63 \pm 0.40$ | $75.31 \pm 0.22$ |

(c) CIFAR-10.

|  | Low Contrast | High Contrast | Original Contrast | Average |
|---|---|---|---|---|
| $\alpha = 10$ | $34.66 \pm 0.47$ | $40.67 \pm 1.29$ | $44.28 \pm 0.84$ | $39.87 \pm 3.96$ |
| $\alpha = 0.1$ | $37.30 \pm 0.73$ | $40.93 \pm 1.54$ | $\mathbf{43.62 \pm 0.91}$ | $40.61 \pm 2.59$ |
| $\alpha = 0.01$ | $\mathbf{39.06 \pm 0.80}$ | $\mathbf{41.13 \pm 1.14}$ | $42.96 \pm 0.91$ | $\mathbf{41.05 \pm 1.59}$ |

Table 5: Worst-case distribution accuracy attained by AD-GDA, DR-DSGD and DRFA.

|  | Fashion-MNIST | CIFAR-10 | COOS7 |
|---|---|---|---|
| AD-GDA | $\mathbf{58.47 \pm 0.92}$ | $\mathbf{39.06 \pm 0.80}$ | $\mathbf{75.00 \pm 0.24}$ |
| DRFA | $56.68 \pm 1.04$ | $37.20 \pm 1.16$ | $69.16 \pm 0.39$ |
| DR-DSGD | $50.77 \pm 0.42$ | $38.00 \pm 0.25$ | $67.09 \pm 0.42$ |

the minimax optimization problem (3), the regularizer $r(\boldsymbol{\lambda})$ reduces the freedom that an adversary has in choosing the weighting vector $\boldsymbol{\lambda}$ to maximize the training loss at every iteration $t$. As a result, the smaller the value of $\alpha$, the less constrained is the adversary and the larger will be the emphasis on the worst-performing nodes. This intuition is confirmed by the following experiments in which we consider a regularizer of the form $\chi^2(\boldsymbol{\lambda}) := \sum_i \frac{(\lambda_i - n_i/n)^2}{n_i/n}$ and run AD-GDA for $\alpha = \{10, 1, 0.01\}$. In Table 4 we report the average accuracy attained in the three simulation setups. For $\alpha = 10$, we observe a large test accuracy gap between the worst and best nodes in the network: 37% for the Fashion-MNIST data set, 9% for the CIFAR-10 data set and 19% for the COOS7 data set. This large accuracy mismatch showcases how large regularization parameter values, and standard decentralized optimization schemes (obtained for $\alpha = \infty$), are unable to guarantee uniform performance across participating parties. On the other hand, using smaller regularization parameters, the gap between is effectively reduced: 31% for the Fashion-MNIST, less than 2% for CIFAR-10 and less than 1% for COOS7. At the same time, the improved fairness brought by AD-GDA does not significantly hamper the average performance of the final model as reported in the last column of all the tables.

In Table 5 we compare the AD-GDA worst-node accuracy against the one obtained by DRFA and DR-DSGD. In particular, we run AD-GDA with a regularization parameter $\alpha = 0.01$ and without message compression. For DR-DSGD we consider the same network setup and we fix the regularization parameter $\alpha = 6$, which we find to yield the best performance. Finally, for DRFA, we organize the network nodes according to a star topology and we run the federated optimization using half-user participation and with a number of local iterations equal to 10. Across all experiments, we find that AD-GDA yields the largest worst-node accuracy. Compared to DR-DSGD, we attribute the superiority of AD-GDA to the capability of solving the distributionally robust optimization using any strongly-concave regularizer, in this case, the chi-squared one. On the other hand, the superiority of DRFA can be attributed to the fact that DRFA attains distributional robustness by sampling more frequently nodes with unsatisfactory performance. However, whenever the fraction of these users is small compared to the overall number of devices that join the federated round, DRFA is unable to prioritize the nodes with the worst performance since they remain under-represent within the group of sampled devices.

### 5.2.2 Communication Efficiency

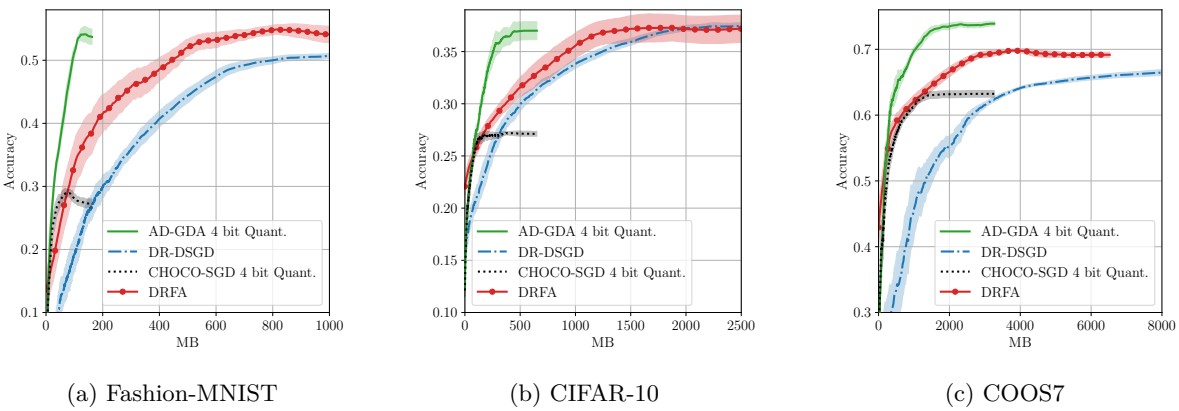

(a) Fashion-MNIST          (b) CIFAR-10          (c) COOS7

Figure 5: Comparison between the proposed algorithm (AD-GDA), Distributionally Robust Federated Averaging (DRFA), Distributionally Robust Decentralized Stochastic Gradient Descent (DR-SGD) and standard decentralized learning (CHOCO-SGD). The algorithms are compared in terms of their communication efficiency and worst node accuracy.

In the following, we compare AD-GDA against CHOCO-SGD and other existing distributionally robust learning schemes; in particular, Distributionally Robust Federated Averaging (DRFA) (Deng et al., 2021) and Distributionally Robust Decentralized Stochastic Gradient Descent (DR-SGD) (Issaid et al., 2022). CHOCO-SGD represents a standard decentralized learning procedure that employs compressed gossip to attain communication efficiency. DRFA is a federated distributionally robust learning scheme with network devices connected according to a star topology and the star center represented by a central aggregator. Communication efficiency is obtained allowing network devices to perform multiple local updates of the primal variable between subsequent synchronization rounds at the central aggregator. We run DRFA allowing devices to perform 10 local gradient steps before sending their local models for the distributionally robust averaging steps and we consider half-user participation at each round. DR-DSGD is a decentralized distributionally robust learning scheme based on KL regularizers that do not employ compressed communication. For the decentralized learning schemes, we consider a 2D torus topology with every node connected to the other four nodes and use Metropolis mixing matrices. For AD-GDA and CHOCO-SGD, which employ compressed communication, we consider a 4-bit quantization scheme. For AD-GDA we consider a chi-squared regularizer with $\alpha = 0.01$, while for DR-DSGD we set the KL regularizer parameter to $\alpha = 6$ as in Issaid et al. (2022). All algorithms are run for $T = 5000$ iterations using an SGD optimizer and the same exponentially learning rate schedule $\eta_\theta^t = r^{-t}\eta_\theta^0$ with decay $r = 0.998$. In order to make the comparison fair, the initial learning rate $\eta_0$ is set differently at every node to ensure that the effective learning rate is equal across different algorithms. In fact, DR-DSGD and AD-GDA descent steps are affected by the dual variable. The batch sizes are the same across all algorithms and are set to $\{50, 50, 32\}$ for the F-MNIST, CIFAR10 and COOS7 experiments, respectively. The algorithms are compared in terms of communication efficiency, namely the capability of producing a fair predictor communicating the smallest number of bits. To this end, in Figure 5 we compare the worst-case distribution accuracy against the number of bits transmitted by the busiest node in the network. The simulation results are reported over the different simulation scenarios. AD-GDA and CHOCO-SGD employ the same compression scheme and converge within the same communication budget. However, AD-GDA can greatly increase the worst node accuracy compared to CHOCO-SGD. This showcases the merits of distributionally robust learning compared to standard learning. Compared to the other distributionally robust algorithms, AD-GDA attains the largest worst-case accuracy by transmitting only a fraction of bits. In particular, for the Fashion-MNIST setup, AD-GDA is 4x and 8x more communication-efficient compared to DRFA and DR-SGD respectively. For the CIFAR-10 data set, AD-GDA reduces by 3x and 5x the number of bits necessary to attain the same final worst-node accuracy of DRFA and DR-SGD respectively. Finally, in the COOS7 case, AD-GDA is 3x more efficient than DRFA and 10x more efficient than DR-SGD.

## 6 Conclusion

We provided a provably convergent decentralized single-loop gradient descent/ascent algorithm to solve the distributionally robust optimization problem over a network of collaborating nodes with heterogeneous local data distributions. Differently from previously proposed solutions, which are either limited to the federated scenario with a central coordinator or to specific regularizers, our algorithm *directly* tackles the underlying minimax optimization problem in a *decentralized* and *communication-efficient* manner. Experiments showed that the proposed solution produces distributionally robust predictors and it attains superior communication efficiency compared to the previously proposed algorithms. A combination of the compressed communication and multiple local updates, combined with acceleration techniques, represents the natural extension of the algorithm to further improve its efficiency.

### Acknowledgments

The work of M. Zecchin is funded by the Marie Curie action WINDMILL (Grant agreement No. 813999). The work of M. Kountouris has received funding from the European Research Council (ERC) under the European Union's Horizon 2020 research and innovation programme (Grant agreement No. 101003431). The work of David Gesbert was partially supported by the 3IA interdisciplinary project ANR-19-P3IA-0002 funded from the French National Research Agency (ANR)"

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

## A   Appendix

### A.1   Useful inequalities

This section contains a collection of ancillary results that are useful for the subsequent proofs.

**Proposition A.1.** *A differentiable and L-smooth function $f(\boldsymbol{x})$ satisfies*

$$\|\nabla f(\boldsymbol{x}) - \nabla f(\boldsymbol{x}')\| \leq L \|\boldsymbol{x} - \boldsymbol{x}'\|. \tag{12}$$

*Furthermore, if $f(\boldsymbol{x})$ is convex*

$$f(\boldsymbol{y}) \leq f(\boldsymbol{x}) + \langle \nabla f(\boldsymbol{x}), \boldsymbol{y} - \boldsymbol{x} \rangle + \frac{L}{2} \|\boldsymbol{y} - \boldsymbol{x}\|^2 \tag{13}$$

*and if $\boldsymbol{x}^*$ is a minimizer*

$$\frac{1}{2L} \|\nabla f(\boldsymbol{x})\|^2 \leq f(\boldsymbol{x}) - f(\boldsymbol{x}^*). \tag{14}$$

*Otherwise, if $f(\boldsymbol{x})$ concave*

$$f(\boldsymbol{y}) \geq f(\boldsymbol{x}) + \langle \nabla f(\boldsymbol{x}), \boldsymbol{y} - \boldsymbol{x} \rangle - \frac{L}{2} \|\boldsymbol{y} - \boldsymbol{x}\|^2 \tag{15}$$

*and if $x^*$ is a maximizer*

$$\frac{1}{2L} \|\nabla f(\boldsymbol{x})\|^2 \leq f(\boldsymbol{x}^*) - f(\boldsymbol{x}). \tag{16}$$

**Proposition A.2.** *A differentiable and $\mu$-strongly convex function $f(\boldsymbol{x})$ satisfies*

$$f(\boldsymbol{y}) \geq f(\boldsymbol{x}) + \langle \nabla f(\boldsymbol{x}), \boldsymbol{y} - \boldsymbol{x} \rangle + \frac{\mu}{2} \|y - \boldsymbol{x}\|^2 \tag{17}$$

*and a differentiable and $\mu$-strongly concave function $g(\boldsymbol{x})$ satisfies*

$$g(\boldsymbol{y}) \leq g(\boldsymbol{x}) + \langle \nabla g(\boldsymbol{x}), \boldsymbol{y} - \boldsymbol{x} \rangle - \frac{\mu}{2} \|y - \boldsymbol{x}\|^2. \tag{18}$$

**Proposition A.3.** *Given two vectors $\boldsymbol{a}, \boldsymbol{b} \in \mathbb{R}^d$, for $\beta > 0$ we have*

$$2\langle \boldsymbol{a}, \boldsymbol{b} \rangle \leq \beta^{-1} \|\boldsymbol{a}\|^2 + \beta \|\boldsymbol{b}\|^2 \tag{19}$$

*and*

$$\|\boldsymbol{a} + \boldsymbol{b}\| \leq (1 + \beta^{-1}) \|\boldsymbol{a}\|^2 + (1 + \beta) \|\boldsymbol{b}\|^2 \tag{20}$$

**Proposition A.4.** *Given two matrices $\boldsymbol{A} \in \mathbb{R}^{p \times q}, \boldsymbol{B} \in \mathbb{R}^{q \times r}$, we have*

$$\|AB\|_F \le \|A\|_F \|B\|_2 \tag{21}$$

*where $\|\cdot\|_F$ denotes the Frobenius norm.*

**Proposition A.5.** *Given a set of vectors $\{\boldsymbol{a}_i\}_{i=1}^n$ we have*

$$\left\| \sum_{i=1}^n \boldsymbol{a}_i \right\|^2 \le n \sum_{i=1}^n \|\boldsymbol{a}_i\|^2. \tag{22}$$

**Consensus inequalities**

To streamline the notation we define $\tilde{\nabla} g_i(\boldsymbol{\theta}_i^t, \boldsymbol{\lambda}_i^t) = \nabla g_i(\boldsymbol{\theta}_i^t, \boldsymbol{\lambda}_i^t, \boldsymbol{\xi}_i^t)$ and introduce the following matrices

$$\Theta^t = \left[\boldsymbol{\theta}_1^t, \ldots, \boldsymbol{\theta}_m^t\right] \in \mathbb{R}^{d \times m}, \quad \hat{\Theta}^t = \left[\hat{\boldsymbol{\theta}}_1^t, \ldots, \hat{\boldsymbol{\theta}}_m^t\right] \in \mathbb{R}^{d \times m}, \quad \Lambda^t = \left[\boldsymbol{\lambda}_1^t, \ldots, \boldsymbol{\lambda}_m^t\right] \in \mathbb{R}^{m \times m}. \tag{23}$$

$$\tilde{\nabla}_{\boldsymbol{\theta}} G(\Theta^t, \Lambda^t) = \left[\tilde{\nabla}_{\boldsymbol{\theta}} g_1(\boldsymbol{\theta}_1^t, \boldsymbol{\lambda}_1^t), \ldots, \tilde{\nabla}_{\boldsymbol{\theta}} g_m(\boldsymbol{\theta}_m^t, \boldsymbol{\lambda}_m^t)\right] \in \mathbb{R}^{d \times m} \tag{24}$$

$$\tilde{\nabla}_{\boldsymbol{\lambda}} G(\Theta^t, \Lambda^t) = \left[\tilde{\nabla}_{\boldsymbol{\lambda}} g_1(\boldsymbol{\theta}_1^t, \boldsymbol{\lambda}_1^t), \ldots, \tilde{\nabla}_{\boldsymbol{\lambda}} g_m(\boldsymbol{\theta}_m^t, \boldsymbol{\lambda}_m^t)\right] \in \mathbb{R}^{m \times m} \tag{25}$$

and for a matrix $X$ we define $\bar{X} = X \frac{\mathbf{1}\mathbf{1}^T}{m}$.

The local update rule of Algorithm 1 can be rewritten as

$$\Theta^{t+\frac{1}{2}} = \Theta^t - \eta_\theta \tilde{\nabla}_{\boldsymbol{\theta}} G(\Theta^t, \Lambda^t) \tag{26}$$

$$\Lambda^{t+\frac{1}{2}} = \mathcal{P}_\Lambda \left(\Lambda^t + \eta_\lambda \tilde{\nabla}_{\boldsymbol{\lambda}} G(\Theta^t, \Lambda^t)\right) \tag{27}$$

where $\mathcal{P}_\Lambda$ is applied column-wise. The compressed gossip algorithm CHOCO-GOSSIP (Koloskova et al., 2019b) used to share model parameters preserves averages and satisfies the following recursive inequality with $c = \frac{\rho^2 \delta}{82}$

$$\mathbb{E}\left[\left\|\Theta^{t+1} - \bar{\Theta}^{t+1}\right\|_F^2 + \left\|\Theta^{t+1} - \hat{\Theta}^{t+1}\right\|_F^2\right] \le (1-c)\,\mathbb{E}\left[\left\|\Theta^{t+\frac{1}{2}} - \bar{\Theta}^{t+\frac{1}{2}}\right\|_F^2 + \left\|\Theta^{t+\frac{1}{2}} - \hat{\Theta}^t\right\|_F^2\right]. \tag{28}$$

The uncompressed gossip scheme used to communicate $\Lambda$ satisfies

$$\mathbb{E}\left[\left\|\Lambda^{t+1} - \bar{\Lambda}^{t+1}\right\|_F^2\right] \le (1-\rho)\,\mathbb{E}\left[\left\|\Lambda^{t+\frac{1}{2}} - \bar{\Lambda}^{t+\frac{1}{2}}\right\|_F^2\right]. \tag{29}$$

**Lemma A.6.** *(Consensus inequality for compressed communication (Koloskova et al., 2019a)) For a fixed $\eta_\theta > 0$ and $\gamma = \frac{\rho^2 \delta}{16\rho + \rho^2 + 4\beta^2 + 2\rho\beta^2 - 8\rho\delta}$ the iterates of Algorithm 1 satisfy*

$$\mathbb{E}\left[\Xi_\theta^t\right] = \mathbb{E}\left[\sum_{i=1}^m \left\|\boldsymbol{\theta}_i^t - \bar{\boldsymbol{\theta}}^t\right\|^2\right] \le 12\eta_\theta^2 \frac{mG_\theta^2}{c^2}. \tag{30}$$

**Lemma A.7.** *(Consensus Inequality for uncompressed communication (Koloskova et al., 2019b)) For a fixed $\eta_\lambda > 0$ the iterates of Algorithm 1 satisfy*

$$\mathbb{E}\left[\Xi_\lambda^t\right] = \mathbb{E}\left[\sum_{i=1}^m \left\|\boldsymbol{\lambda}_i^t - \bar{\boldsymbol{\lambda}}^t\right\|^2\right] \le 4\eta_\lambda^2 \frac{mG_\lambda^2}{\rho^2} \tag{31}$$

Lemma A.6 and A.7 apply to the primal and dual iterates obtained of Algorithm 1 and they are obtained from Lemma A.2 of (Koloskova et al., 2019b).

## A.2 Proof of Theorem 4.1: Convex case

Define

$$\Phi(\cdot) = \max_{\boldsymbol{\lambda} \in \Delta^{m-1}} g(\cdot, \boldsymbol{\lambda}); \tag{32}$$

under assumptions 3.3, 3.4 and if the local objective functions $\{f_i(\boldsymbol{\theta})\}_{i=1}^m$ are convex, Theorem 4.1 guarantees that the output solution $(\boldsymbol{\theta}_o, \boldsymbol{\lambda}_o)$ satisfies

$$\mathbb{E}\left[\Phi(\boldsymbol{\theta}_o) - \min_{\boldsymbol{\theta} \in \Theta} \Phi(\boldsymbol{\theta})\right] \leq \frac{4}{T}\left(\frac{LG_\lambda^2}{\rho^2} + 3\frac{LG_\theta^2}{c^2}\right) + \frac{1}{\sqrt{T}}\left(\sqrt{12}\frac{D_\lambda LG_\theta}{c} + 2\frac{D_\theta LG_\lambda}{\rho}\right)$$
$$+ \frac{1}{\sqrt{T}}\left(\frac{D_\theta + D_\lambda}{2} + \frac{G_\theta^2 + G_\lambda^2}{2}\right). \tag{33}$$

The proof starts from the following decomposition of the sub-optimality gap

$$\mathbb{E}\left[\max_{\boldsymbol{\lambda}} g(\boldsymbol{\theta}_o, \boldsymbol{\lambda}) - \min_{\boldsymbol{\theta}} \max_{\boldsymbol{\lambda}} g(\boldsymbol{\theta}, \boldsymbol{\lambda})\right] \leq \mathbb{E}\left[\max_{\boldsymbol{\lambda}} g(\boldsymbol{\theta}_o, \boldsymbol{\lambda}) - \max_{\boldsymbol{\lambda}} \min_{\boldsymbol{\theta}} g(\boldsymbol{\theta}, \boldsymbol{\lambda}))\right] \tag{34}$$

$$\leq \mathbb{E}\left[\max_{\boldsymbol{\lambda}} g(\boldsymbol{\theta}_o, \boldsymbol{\lambda}) - \min_{\boldsymbol{\theta}} g(\boldsymbol{\theta}, \boldsymbol{\lambda}_o))\right] \tag{35}$$

$$\leq \mathbb{E}\left[\max_{\boldsymbol{\lambda}, \boldsymbol{\theta}} g(\boldsymbol{\theta}_o, \boldsymbol{\lambda}) - g(\boldsymbol{\theta}, \boldsymbol{\lambda}_o))\right] \tag{36}$$

$$\leq \mathbb{E}\left[\max_{\boldsymbol{\lambda}, \boldsymbol{\theta}} \frac{1}{T}\sum_{t=0}^{T-1} g(\bar{\boldsymbol{\theta}}^t, \boldsymbol{\lambda}) - g(\boldsymbol{\theta}, \bar{\boldsymbol{\lambda}}^t))\right] \tag{37}$$

$$\leq \mathbb{E}\left[\max_{\boldsymbol{\lambda}} \frac{1}{T}\sum_{t=0}^{T-1} g(\bar{\boldsymbol{\theta}}^t, \boldsymbol{\lambda}) - g(\bar{\boldsymbol{\theta}}^t, \bar{\boldsymbol{\lambda}}^t)\right]$$
$$+ \mathbb{E}\left[\max_{\boldsymbol{\theta}} \frac{1}{T}\sum_{t=0}^{T-1} g(\bar{\boldsymbol{\theta}}^t, \bar{\boldsymbol{\lambda}}^t) - g(\boldsymbol{\theta}, \bar{\boldsymbol{\lambda}}^t)\right]. \tag{38}$$

Thanks to Lemmas (A.8) and (A.9) proved below, the two summands can be bounded to obtain

$$\mathbb{E}\left[\Phi(\boldsymbol{\theta}_o) - \min_{\boldsymbol{\theta} \in \Theta} \Phi(\boldsymbol{\theta})\right] \leq \frac{D_\theta}{2\eta_\theta T} + \frac{\eta_\theta}{2}\left(G_\theta^2 + \sqrt{48}\frac{D_\lambda LG_\theta}{c}\right) + 12\eta_\theta^2 \frac{LG_\theta^2}{c^2}$$
$$+ \frac{D_\lambda}{2\eta_\lambda T} + \frac{\eta_\lambda}{2}\left(G_\lambda^2 + 4\frac{D_\theta LG_\lambda}{\delta}\right) + 4\eta_\lambda^2 \frac{LG_\lambda^2}{\rho^2}. \tag{39}$$

Setting $\eta_\lambda = \eta_\theta = \frac{1}{\sqrt{T}}$, the final result is obtained. □

**Lemma A.8.** *For $T > 0$ and any $\boldsymbol{\theta}$, the sequence $\{\bar{\boldsymbol{\theta}}^t, \bar{\boldsymbol{\lambda}}^t\}_{t=0}^T$ generated by Algorithm 1 satisfies*

$$\mathbb{E}\left[\frac{1}{T}\sum_{t=0}^{T-1} g(\bar{\boldsymbol{\theta}}^t, \bar{\boldsymbol{\lambda}}^t) - g(\boldsymbol{\theta}, \bar{\boldsymbol{\lambda}}^t)\right] \leq \frac{D_\theta}{2\eta_\theta T} + \frac{\eta_\theta}{2}G_\theta^2 + 12\eta_\theta^2 \frac{LG_\theta^2}{c^2} + 2\eta_\lambda \frac{D_\theta LG_\lambda}{\rho} \tag{40}$$

*where $D_\theta = \max_{t=0...,T} \mathbb{E}\left\|\bar{\boldsymbol{\theta}}^t - \boldsymbol{\theta}\right\|$.*

**Proof:** From the update rule of the primal variable and the assumptions 3.4 on the stochastic gradient we have, that for any $\boldsymbol{\theta}$

$$\mathbb{E}_{\boldsymbol{\xi}^t}\left\|\bar{\boldsymbol{\theta}}^{t+1} - \boldsymbol{\theta}\right\|^2 = \mathbb{E}_{\boldsymbol{\xi}^t}\left\|\bar{\boldsymbol{\theta}}^t - \boldsymbol{\theta} - \frac{\eta_\theta}{m}\sum_{i=1}^m \tilde{\nabla}_{\boldsymbol{\theta}} g_i(\boldsymbol{\theta}_i^t, \boldsymbol{\lambda}_i^t)\right\|^2 \tag{41}$$

$$= \left\|\bar{\boldsymbol{\theta}}^t - \boldsymbol{\theta}\right\|^2 - 2\frac{\eta_\theta}{m}\sum_{i=1}^m \langle \bar{\boldsymbol{\theta}}^t - \boldsymbol{\theta}; \mathbb{E}_{\boldsymbol{\xi}^t}\left[\tilde{\nabla}_{\boldsymbol{\theta}} g_i(\boldsymbol{\theta}_i^t, \boldsymbol{\lambda}_i^t)\right]\rangle + \mathbb{E}_{\boldsymbol{\xi}^t}\left\|\frac{\eta_\theta}{m}\sum_{i=1}^m \tilde{\nabla}_{\boldsymbol{\theta}} g_i(\boldsymbol{\theta}_i^t, \boldsymbol{\lambda}_i^t)\right\|^2 \tag{42}$$

$$\leq \left\|\bar{\boldsymbol{\theta}}^t - \boldsymbol{\theta}\right\|^2 \underbrace{-2\frac{\eta_\theta}{m}\sum_{i=1}^m \langle \bar{\boldsymbol{\theta}}^t - \boldsymbol{\theta}; \nabla_{\boldsymbol{\theta}} g_i(\boldsymbol{\theta}_i^t, \boldsymbol{\lambda}_i^t)\rangle}_{:=T_2} + \eta_\theta^2 G_\theta^2. \tag{43}$$

Denoting with $D_\theta^t = \left\|\bar{\boldsymbol{\theta}}^t - \boldsymbol{\theta}\right\|$ we have that for $T_2$ the following holds

$$T_2 = -2\frac{\eta_\theta}{m}\left(\sum_{i=1}^m \langle \bar{\boldsymbol{\theta}}^t - \boldsymbol{\theta}; \nabla_{\boldsymbol{\theta}} g_i(\boldsymbol{\theta}_i^t, \bar{\boldsymbol{\lambda}}^t)\rangle + \sum_{i=1}^m \langle \bar{\boldsymbol{\theta}}^t - \boldsymbol{\theta}; \nabla_{\boldsymbol{\theta}} g_i(\boldsymbol{\theta}_i^t, \boldsymbol{\lambda}_i^t) - \nabla_{\boldsymbol{\theta}} g_i(\boldsymbol{\theta}_i^t, \bar{\boldsymbol{\lambda}}^t)\rangle\right) \tag{44}$$

$$\leq -2\frac{\eta_\theta}{m}\sum_{i=1}^m \langle \bar{\boldsymbol{\theta}}^t - \boldsymbol{\theta}; \nabla_{\boldsymbol{\theta}} g_i(\boldsymbol{\theta}_i^t, \bar{\boldsymbol{\lambda}}^t)\rangle + 2\eta_\theta L D_\theta^t \sqrt{\frac{\Xi_\lambda^t}{m}} \tag{45}$$

$$\leq -2\frac{\eta_\theta}{m}\sum_{i=1}^m \left(\langle \bar{\boldsymbol{\theta}}^t - \boldsymbol{\theta}_i^t; \nabla_{\boldsymbol{\theta}} g_i(\boldsymbol{\theta}_i^t, \bar{\boldsymbol{\lambda}}^t)\rangle + \langle \boldsymbol{\theta}_i^t - \boldsymbol{\theta}; \nabla_{\boldsymbol{\theta}} g_i(\boldsymbol{\theta}_i^t, \bar{\boldsymbol{\lambda}}^t)\rangle\right) + 2\eta_\theta L D_\theta^t \sqrt{\frac{\Xi_\lambda^t}{m}} \tag{46}$$

$$\overset{(15)}{\leq} -2\frac{\eta_\theta}{m}\sum_{i=1}^m \left(g_i(\bar{\boldsymbol{\theta}}^t, \bar{\boldsymbol{\lambda}}^t) - g_i(\boldsymbol{\theta}, \bar{\boldsymbol{\lambda}}^t) - \frac{L}{2}\left\|\bar{\boldsymbol{\theta}}^t - \boldsymbol{\theta}_i^t\right\|^2\right) + 2\eta_\theta L D_\theta^t \sqrt{\frac{\Xi_\lambda^t}{m}} \tag{47}$$

$$= -2\frac{\eta_\theta}{m}\sum_{i=1}^m \left(g_i(\bar{\boldsymbol{\theta}}^t, \bar{\boldsymbol{\lambda}}^t) - g_i(\boldsymbol{\theta}, \bar{\boldsymbol{\lambda}}^t)\right) + \frac{2\eta_\theta L}{m}\Xi_\theta^t + 2\eta_\theta L D_\theta^t \sqrt{\frac{\Xi_\lambda^t}{m}}. \tag{48}$$

Plugging it back in (43), rearranging the terms and taking the expectation over the previous iterate we get

$$\mathbb{E}\left[g(\bar{\boldsymbol{\theta}}^t, \bar{\boldsymbol{\lambda}}^t) - g(\boldsymbol{\theta}, \bar{\boldsymbol{\lambda}}^t)\right] = \frac{1}{m}\mathbb{E}\left[\sum_{i=1}^m g_i(\bar{\boldsymbol{\theta}}^t, \bar{\boldsymbol{\lambda}}^t) - g_i(\boldsymbol{\theta}, \bar{\boldsymbol{\lambda}}^t)\right] \tag{49}$$

$$\leq \frac{\mathbb{E}\|\bar{\boldsymbol{\theta}}^t - \boldsymbol{\theta}\|^2 - \mathbb{E}\|\bar{\boldsymbol{\theta}}^{t+1} - \boldsymbol{\theta}\|^2}{2\eta_\theta} + \frac{\eta_\theta}{2}G_\theta^2 + \frac{L}{m}\mathbb{E}\left[\Xi_\theta^t\right] + L\mathbb{E}\left[D_\theta^t\right]\sqrt{\frac{\mathbb{E}\left[\Xi_\lambda^t\right]}{m}}. \tag{50}$$

Telescoping from $t = 0$ to $t = T - 1$ and plugging the consensus inequalities (30) and (31), we get

$$\frac{1}{T}\mathbb{E}\left[\sum_{t=0}^{T-1} g(\bar{\boldsymbol{\theta}}^t, \bar{\boldsymbol{\lambda}}^t) - g(\boldsymbol{\theta}, \bar{\boldsymbol{\lambda}}^t)\right] \leq \frac{D_\theta}{2\eta_\theta T} + \frac{\eta_\theta}{2}G_\theta^2 + 12\eta_\theta^2\frac{LG_\theta^2}{c^2} + 2\eta_\lambda\frac{D_\theta LG_\lambda}{\rho} \tag{51}$$

where $D_\theta = \max_{t=0...,T}\mathbb{E}[D_\theta^t] = \max_{t=0...,T}\mathbb{E}\left\|\bar{\boldsymbol{\theta}}^t - \boldsymbol{\theta}\right\|$. $\qquad\square$

**Lemma A.9.** *For $T > 0$ and any $\boldsymbol{\lambda}$, the sequence $\{\bar{\boldsymbol{\theta}}^t, \bar{\boldsymbol{\lambda}}^t\}_{t=0}^T$ generated by Algorithm 1 satisfies*

$$\mathbb{E}\left[\frac{1}{T}\sum_{t=0}^{T-1} g(\bar{\boldsymbol{\theta}}^t, \boldsymbol{\lambda}) - g(\bar{\boldsymbol{\theta}}^t, \bar{\boldsymbol{\lambda}}^t)\right] \leq \frac{D_\lambda}{2\eta_\lambda T} + \frac{\eta_\lambda}{2}G_\lambda^2 + 4\eta_\lambda^2\frac{LG_\lambda^2}{\rho^2} + \sqrt{12}\eta_\theta\frac{D_\lambda LG_\theta}{c} \tag{52}$$

*where $D_\lambda = \max_{t=0...,T}\mathbb{E}\left\|\bar{\boldsymbol{\lambda}}^t - \boldsymbol{\lambda}\right\|$.*

**Proof** The proof follows similarly as in Lemma (A.8)

$$\mathbb{E}_{\boldsymbol{\xi}^t}\left\|\bar{\boldsymbol{\lambda}}^{t+1} - \boldsymbol{\lambda}\right\|^2 = \mathbb{E}_{\boldsymbol{\xi}^t}\left\|\boldsymbol{\lambda} - \bar{\boldsymbol{\lambda}}^t + \frac{\eta_\lambda}{m}\sum_{i=1}^m \tilde{\nabla}_{\boldsymbol{\lambda}} g_i(\boldsymbol{\theta}_i^t, \boldsymbol{\lambda}_i^t)\right\|^2 \tag{53}$$

$$= \left\|\bar{\boldsymbol{\lambda}}^t - \boldsymbol{\lambda}\right\|^2 - 2\frac{\eta_\lambda}{m}\sum_{i=1}^m \langle \boldsymbol{\lambda} - \bar{\boldsymbol{\lambda}}^t; \mathbb{E}_{\boldsymbol{\xi}^t}\left[\tilde{\nabla}_{\boldsymbol{\lambda}} g_i(\boldsymbol{\theta}_i^t, \boldsymbol{\lambda}_i^t)\right]\rangle + \mathbb{E}_{\boldsymbol{\xi}^t}\left\|\frac{\eta_\lambda}{m}\sum_{i=1}^m \tilde{\nabla}_{\boldsymbol{\lambda}} g_i(\boldsymbol{\theta}_i^t, \boldsymbol{\lambda}_i^t)\right\|^2 \tag{54}$$

$$= \mathbb{E}\left\|\bar{\boldsymbol{\lambda}}^t - \boldsymbol{\lambda}\right\|^2 \underbrace{-2\frac{\eta_\lambda}{m}\sum_{i=1}^m \langle \boldsymbol{\lambda} - \bar{\boldsymbol{\lambda}}^t; \nabla_{\boldsymbol{\lambda}} g_i(\boldsymbol{\theta}_i^t, \boldsymbol{\lambda}_i^t)\rangle}_{:=T_3} + \eta_\lambda^2 G_\lambda^2. \tag{55}$$

Denoting with $D_\lambda^t = \left\| \bar{\boldsymbol{\lambda}}^t - \boldsymbol{\lambda} \right\|$ we have that for $T_3$ the following holds

$$T_3 = -2\frac{\eta_\lambda}{m} \left( \sum_{i=1}^m \langle \boldsymbol{\lambda} - \bar{\boldsymbol{\lambda}}^t ; \nabla_{\boldsymbol{\lambda}} g_i(\bar{\boldsymbol{\theta}}^t, \boldsymbol{\lambda}_i^t) \rangle + \sum_{i=1}^m \langle \boldsymbol{\lambda} - \bar{\boldsymbol{\lambda}}^t ; \nabla_{\boldsymbol{\lambda}} g_i(\boldsymbol{\theta}_i^t, \boldsymbol{\lambda}_i^t) - \nabla_{\boldsymbol{\lambda}} g_i(\bar{\boldsymbol{\theta}}^t, \boldsymbol{\lambda}_i^t) \rangle \right) \tag{56}$$

$$\leq -2\frac{\eta_\lambda}{m} \sum_{i=1}^m \left( \langle \boldsymbol{\lambda} - \bar{\boldsymbol{\lambda}}^t ; \nabla_{\boldsymbol{\lambda}} g_i(\bar{\boldsymbol{\theta}}^t, \boldsymbol{\lambda}_i^t) \rangle \right) + 2\eta_\lambda L D_\lambda^t \sqrt{\frac{\Xi_\theta^t}{m}} \tag{57}$$

$$\leq -2\frac{\eta_\lambda}{m} \sum_{i=1}^m \left( \langle \boldsymbol{\lambda} - \boldsymbol{\lambda}_i^t ; \nabla_{\boldsymbol{\lambda}} g_i(\bar{\boldsymbol{\theta}}^t, \boldsymbol{\lambda}_i^t) \rangle + \langle \boldsymbol{\lambda}_i^t - \bar{\boldsymbol{\lambda}}^t ; \nabla_{\boldsymbol{\lambda}} g_i(\bar{\boldsymbol{\theta}}^t, \boldsymbol{\lambda}_i^t) \rangle \right) + 2\eta_\lambda L D_\lambda^t \sqrt{\frac{\Xi_\theta^t}{m}} \tag{58}$$

$$\overset{(15)}{\leq} -2\frac{\eta_\lambda}{m} \sum_{i=1}^m \left( g_i(\bar{\boldsymbol{\theta}}^t, \boldsymbol{\lambda}) - g_i(\bar{\boldsymbol{\theta}}^t, \bar{\boldsymbol{\lambda}}^t) - \frac{L}{2} \left\| \bar{\boldsymbol{\lambda}}^t - \boldsymbol{\lambda}_i^t \right\|^2 \right) + 2\eta_\lambda L D_\lambda^t \sqrt{\frac{\Xi_\theta^t}{m}} \tag{59}$$

$$= -2\frac{\eta_\lambda}{m} \sum_{i=1}^m \left( g_i(\bar{\boldsymbol{\theta}}^t, \boldsymbol{\lambda}) - g_i(\bar{\boldsymbol{\theta}}^t, \bar{\boldsymbol{\lambda}}^t) \right) + \frac{2\eta_\lambda L}{m} \Xi_\lambda^t + 2\eta_\lambda L D_\lambda^t \sqrt{\frac{\Xi_\theta^t}{m}}. \tag{60}$$

Plugging it back in (55), rearranging the terms and taking the expectation over the previous iterate we get

$$\mathbb{E} \left[ g(\bar{\boldsymbol{\theta}}^t, \boldsymbol{\lambda}) - g(\bar{\boldsymbol{\theta}}^t, \bar{\boldsymbol{\lambda}}^t) \right] = \frac{1}{m} \mathbb{E} \left[ \sum_{i=1}^m g_i(\bar{\boldsymbol{\theta}}^t, \boldsymbol{\lambda}) - g_i(\bar{\boldsymbol{\theta}}^t, \bar{\boldsymbol{\lambda}}^t) \right] \tag{61}$$

$$\leq \frac{\mathbb{E}\|\bar{\boldsymbol{\lambda}}^t - \boldsymbol{\lambda}\|^2 - \mathbb{E}\|\bar{\boldsymbol{\lambda}}^{t+1} - \boldsymbol{\lambda}\|^2}{2\eta_\lambda} + \frac{\eta_\lambda}{2} G_\lambda^2 + \frac{L}{m} \mathbb{E} \left[ \Xi_\lambda^t \right] + L\mathbb{E} \left[ D_\lambda^t \right] \sqrt{\frac{\mathbb{E} \left[ \Xi_\theta^t \right]}{m}}. \tag{62}$$

Telescoping from $t = 0$ to $t = T - 1$ and plugging the consensus inequalities (30) and (31) we get

$$\frac{1}{T} \mathbb{E} \left[ \sum_{t=0}^{T-1} g(\bar{\boldsymbol{\theta}}^t, \boldsymbol{\lambda}) - g(\bar{\boldsymbol{\theta}}^t, \bar{\boldsymbol{\lambda}}^t) \right] \leq \frac{D_\lambda}{2\eta_\lambda T} + \frac{\eta_\lambda}{2} G_\lambda^2 + 4\eta_\lambda^2 \frac{LG_\lambda^2}{\rho^2} + \sqrt{12}\eta_\theta \frac{D_\lambda L G_\theta}{c} \tag{63}$$

where $D_\lambda = \max_{t=0\ldots,T} \mathbb{E}[D_\lambda^t] = \max_{t=0\ldots,T} \mathbb{E} \left\| \bar{\boldsymbol{\lambda}}^t - \boldsymbol{\lambda} \right\|$. $\qquad\qquad\square$

### A.3 Proof of Theorem 4.3: Non-convex case

In the case of non-convex functions $\{f_i\}_{i=1}^m$, Theorem 4.3 provides the following $\epsilon$-stationarity guarantee on the randomized solution of Algorithm 1 :

$$\frac{1}{T} \sum_{t=1}^T \mathbb{E} \left[ \left\| \nabla \Phi(\bar{\boldsymbol{\theta}}^{t-1}) \right\|^2 \right] \leq \frac{2L}{\sqrt{T}} \left( 256 \left( \mathbb{E}[\Phi(\bar{\boldsymbol{\theta}}^0)] - \mathbb{E}[\Phi(\bar{\boldsymbol{\theta}}^T)] \right) + \frac{45L\kappa^2 D_\lambda^0}{2} \right)$$
$$+ \frac{1}{\sqrt{T}} \left( 5D_\lambda L \frac{G_\theta}{c} + \frac{\sigma_\theta^2}{2m} + \frac{45\kappa\sigma_\lambda^2}{4m} \right) + \frac{1}{T} \left( \frac{G_\theta^2}{4c^2} + 171\frac{\kappa G_\lambda^2}{\rho^2} \right) + \frac{\sigma_\theta^2}{m}. \tag{64}$$

The proof is inspired from recent results in (Lin et al., 2020a). Specifically, Lemma A.10, stated and proved below, provides a descent inequality of the type

$$\mathbb{E}[\Phi(\bar{\boldsymbol{\theta}}^t)] \leq \mathbb{E}[\Phi(\bar{\boldsymbol{\theta}}^{t-1})] + \frac{\eta_\theta^2 \kappa L \sigma_\theta^2}{m} - \left( \frac{\eta_\theta}{2} - 2\eta_\theta^2 \kappa L \right) \mathbb{E} \left[ \nabla \| \Phi(\bar{\boldsymbol{\theta}}^{t-1}) \|^2 \right]$$
$$+ L^2 \left( \frac{\eta_\theta}{2} + 2\eta_\theta^2 \kappa L \right) \left( \frac{\mathbb{E}[\Xi_\theta^t]}{m} + \frac{2\mathbb{E}[\Xi_\lambda^t]}{m} + 2\mathbb{E}[\delta_\lambda^t] \right). \tag{65}$$

Setting $\eta_\theta = \frac{\eta_\lambda}{16(\kappa+1)^2}$ and $\eta_\lambda \leq \frac{1}{2L}$ expression (65) can be simplified thanks to the following chain of inequalities

$$\frac{7\eta_\theta}{16} \leq \eta_\theta(\frac{1}{2} - 2\eta_\theta \kappa L) \leq \eta_\theta(\frac{1}{2} + 2\eta_\theta \kappa L) \leq \frac{9\eta_\theta}{16}. \tag{66}$$

Telescoping the simplified expression from $t = 1$ to $T$ we obtain

$$\mathbb{E}[\Phi(\bar{\boldsymbol{\theta}}^T)] \leq \mathbb{E}[\Phi(\bar{\boldsymbol{\theta}}^0)] + T\frac{\eta_\theta^2 \kappa L \sigma_\theta^2}{m} - \frac{7\eta_\theta}{16} \sum_{t=1}^{T} \mathbb{E}\left[\nabla \|\Phi(\bar{\boldsymbol{\theta}}^{t-1})\|^2\right]$$
$$+ L^2 \frac{9\eta_\theta}{16} \sum_{t=1}^{T} \left(\frac{\mathbb{E}[\Xi_\theta^t]}{m} + \frac{2\mathbb{E}[\Xi_\lambda^t]}{m}\right) + \frac{9\eta_\theta L^2}{8} \mathbb{E}\left[\sum_{t=1}^{T} \delta_\lambda^t\right] \tag{67}$$

where $\delta_\lambda^t := \|\boldsymbol{\lambda}^*(\bar{\boldsymbol{\theta}}^t) - \bar{\boldsymbol{\lambda}}^t\|^2$ represents the squared distance between the optimal value of the dual variable for the current averaged network belief and the current averaged value of the dual variable.

Lemma A.11, reported below, provides a bound on $\sum_{t=1}^{T} \delta_\lambda^t$ that plugged in (67) yields

$$\mathbb{E}[\Phi(\bar{\boldsymbol{\theta}}^T)] \leq \mathbb{E}[\Phi(\bar{\boldsymbol{\theta}}^0)] + \eta_\theta \frac{45 L \kappa^2 \delta_\lambda^0}{8\eta_\lambda} + \eta_\theta \left(\frac{45\kappa^4 \eta_\theta^2}{\eta_\lambda^2} - \frac{7}{16}\right) \sum_{t=1}^{T} \mathbb{E}\left[\nabla \|\Phi(\bar{\boldsymbol{\theta}}^{t-1})\|^2\right]$$
$$+ T\eta_\theta \left(\frac{\eta_\theta \kappa L \sigma_\theta^2}{m} + \frac{45\kappa L \eta_\lambda \sigma_\lambda^2}{4m} + \frac{45\sigma_\theta^2}{2 \cdot 16^2 m}\right)$$
$$+ L^2 \frac{9\eta_\theta}{16} \sum_{t=1}^{T} \left(\frac{\mathbb{E}[\Xi_\theta^t]}{m} + \frac{2\mathbb{E}[\Xi^t]_\lambda}{m} + \frac{30\kappa\mathbb{E}[\Xi_\theta^{t-1}]}{m} + \frac{70\kappa\mathbb{E}[\Xi_\lambda^{t-1}]}{m}\right)$$
$$+ L^2 \frac{9\eta_\theta}{16} \sum_{t=1}^{T} \left(40\kappa D_\lambda^{t-1} \sqrt{\frac{1}{m}\mathbb{E}[\Xi_\theta^{t-1}]}\right). \tag{68}$$

Moreover, the relation between the two step-sizes established above ensures that

$$\left(\frac{45\kappa^4 \eta_\theta^2}{\eta_\lambda^2} - \frac{7}{16}\right) \leq -\frac{1}{4} \tag{69}$$

and therefore rearranging terms, dividing by $\frac{4}{T\eta_\theta}$ and recalling that $\kappa \geq 1$

$$\frac{1}{T} \sum_{t=1}^{T} \mathbb{E}\left[\nabla \|\Phi(\bar{\boldsymbol{\theta}}^{t-1})\|^2\right] \leq \frac{4}{\eta_\theta T}\left(\mathbb{E}[\Phi(\bar{\boldsymbol{\theta}}^0)] - \mathbb{E}[\Phi(\bar{\boldsymbol{\theta}}^T)]\right) + \frac{45 L \kappa^2 \delta_\lambda^0}{2T\eta_\lambda}$$
$$+ 4\left(\frac{\eta_\theta \kappa L \sigma_\theta^2}{m} + \frac{45\kappa L \eta_\lambda \sigma_\lambda^2}{4m} + \frac{45\sigma_\theta^2}{2 \cdot 16^2 m} +\right)$$
$$+ \frac{9L^2}{4T} \sum_{t=1}^{T} \left(\frac{31\kappa\mathbb{E}[\Xi_\theta^{t-1}]}{m} + \frac{72\mathbb{E}[\kappa\Xi_\lambda^{t-1}]}{m} + \frac{31\kappa\mathbb{E}[\Xi_\theta^{t-1}]}{m} + \frac{72\mathbb{E}[\kappa\Xi_\lambda^{t-1}]}{m}\right). \tag{70}$$

Exploiting consensus inequalities (30), (31) and the fact that $\kappa \geq 1$ and $\eta_\theta = \frac{\eta_\lambda}{16(\kappa+1)^2} \leq 1/2L$ we can simplify and obtain

$$\frac{1}{T} \sum_{t=1}^{T} \mathbb{E}\left[\nabla \|\Phi(\bar{\boldsymbol{\theta}}^{t-1})\|^2\right] \leq \frac{64(\kappa+1)^2}{\eta_\lambda T}\left(\mathbb{E}[\Phi(\bar{\boldsymbol{\theta}}^0)] - \mathbb{E}[\Phi(\bar{\boldsymbol{\theta}}^T)]\right) + \frac{45 L \kappa^2 \delta_\lambda^0}{2T\eta_\lambda}$$
$$+ 2\left(\eta_\lambda \frac{L\sigma_\theta^2}{m} + \eta_\lambda \frac{45\kappa L \sigma_\lambda^2}{2m} + \frac{45\sigma_\theta^2}{16^2 m}\right)$$
$$+ L^2 \frac{9}{4T} \sum_{t=1}^{T} \left(40\kappa D_\lambda^{t-1} \sqrt{12}\eta_\theta \frac{G_\theta}{c} + 372\eta_\theta^2 \frac{\kappa G_\theta^2}{c^2} + 288\eta_\lambda^2 \frac{\kappa G_\lambda^2}{\rho^2}\right). \tag{71}$$

Simplifying and defining $D_\lambda = \max_{t=0,\dots,T} D_\lambda^t$

$$\frac{1}{T} \sum_{t=1}^{T} \mathbb{E}\left[\nabla \|\Phi(\bar{\boldsymbol{\theta}}^{t-1})\|^2\right] \leq \frac{64(\kappa+1)^2}{\eta_\lambda T}\left(\mathbb{E}[\Phi(\bar{\boldsymbol{\theta}}^0)] - \mathbb{E}[\Phi(\bar{\boldsymbol{\theta}}^T)]\right) + \frac{45 L \kappa^2 \delta_\lambda^0}{2T\eta_\lambda}$$

$$+ 2 \left( \eta_\lambda \frac{L\sigma_\theta^2}{m} + \eta_\lambda \frac{45\kappa L\sigma_\lambda^2}{2m} + \frac{45\sigma_\theta^2}{16^2 m} \right)$$

$$+ L^2 \left( 10 D_\lambda \eta_\lambda \frac{G_\theta}{c} + \eta_\lambda^2 \frac{G_\theta^2}{c^2} + 684\eta_\lambda^2 \frac{\kappa G_\lambda^2}{\rho^2} \right). \tag{72}$$

Grouping

$$\frac{1}{T} \sum_{t=1}^{T} \mathbb{E} \left[ \nabla \left\| \Phi(\bar{\boldsymbol{\theta}}^{t-1}) \right\|^2 \right] \leq \frac{1}{\eta_\lambda T} \left( 256 \left( \mathbb{E}[\Phi(\bar{\boldsymbol{\theta}}^0)] - \mathbb{E}[\Phi(\bar{\boldsymbol{\theta}}^T)] \right) + \frac{45 L\kappa^2 \delta_\lambda^0}{2} \right)$$

$$+ \eta_\lambda \left( 10 D_\lambda L^2 \frac{G_\theta}{c} + \frac{L\sigma_\theta^2}{m} + \frac{45\kappa L\sigma_\lambda^2}{2m} \right) + \eta_\lambda^2 \left( \frac{L^2 G_\theta^2}{c^2} + 684 \frac{L^2 \kappa G_\lambda^2}{\rho^2} \right) + \frac{\sigma_\theta^2}{m}. \tag{73}$$

Setting $\eta_\lambda = \frac{1}{2L\sqrt{T}}$ we get

$$\frac{1}{T} \sum_{t=1}^{T} \mathbb{E} \left[ \nabla \left\| \Phi(\bar{\boldsymbol{\theta}}^{t-1}) \right\|^2 \right] \leq \frac{2L}{\sqrt{T}} \left( 256 \left( \mathbb{E}[\Phi(\bar{\boldsymbol{\theta}}^0)] - \mathbb{E}[\Phi(\bar{\boldsymbol{\theta}}^T)] \right) + \frac{45 L\kappa^2 \delta_\lambda^0}{2} \right)$$

$$+ \frac{1}{\sqrt{T}} \left( 5 D_\lambda L \frac{G_\theta}{c} + \frac{\sigma_\theta^2}{2m} + \frac{45\kappa\sigma_\lambda^2}{4m} \right) + \frac{1}{T} \left( \frac{G_\theta^2}{4c^2} + 171 \frac{\kappa G_\lambda^2}{\rho^2} \right) + \frac{\sigma_\theta^2}{m}. \tag{74}$$

**Lemma A.10.** *For each $t = 1, \ldots, T$ the iterates generated by Algorithm 1 satisfies*

$$\mathbb{E}[\Phi(\bar{\boldsymbol{\theta}}^t)] \leq \mathbb{E}[\Phi(\bar{\boldsymbol{\theta}}^{t-1})] + \frac{\eta_\theta^2 \kappa L\sigma_\theta^2}{m} - \left( \frac{\eta_\theta}{2} - 2\eta_\theta^2 \kappa L \right) \mathbb{E} \left[ \nabla \| \Phi(\bar{\boldsymbol{\theta}}^{t-1}) \|^2 \right]$$

$$+ L^2 \left( \frac{\eta_\theta}{2} + 2\eta_\theta^2 \kappa L \right) \left( \frac{\mathbb{E}[\Xi_\theta^t]}{m} + \frac{2\mathbb{E}[\Xi_\lambda^t]}{m} + 2\mathbb{E}[\delta_\lambda^t] \right). \tag{75}$$

**Proof:** From the $2\kappa L$-smoothness of $\Phi(\cdot)$ (Lemma 4.3 of Lin et al. (2020a)) and the update rule we have:

$$\mathbb{E}_{\boldsymbol{\xi}^{t-1}} \left[ \Phi(\bar{\boldsymbol{\theta}}^t) \right] \leq \Phi(\bar{\boldsymbol{\theta}}^{t-1}) + \mathbb{E}_{\boldsymbol{\xi}^{t-1}} \left[ \langle \nabla_{\boldsymbol{\theta}} \Phi(\bar{\boldsymbol{\theta}}^{t-1}), \bar{\boldsymbol{\theta}}^t - \bar{\boldsymbol{\theta}}^{t-1} \rangle \right] + \kappa L \mathbb{E}_{\boldsymbol{\xi}^{t-1}} \left\| \bar{\boldsymbol{\theta}}^t - \bar{\boldsymbol{\theta}}^{t-1} \right\|^2 \tag{76}$$

$$\leq \Phi(\bar{\boldsymbol{\theta}}^{t-1}) - \eta_\theta \langle \nabla_{\boldsymbol{\theta}} \Phi(\bar{\boldsymbol{\theta}}^{t-1}), \frac{1}{m} \sum_{i=1}^{m} \mathbb{E}_{\boldsymbol{\xi}^{t-1}} \left[ \tilde{\nabla}_{\boldsymbol{\theta}} g_i(\boldsymbol{\theta}_i^{t-1}, \boldsymbol{\lambda}_i^{t-1}) \right] \rangle$$

$$+ \frac{\eta_\theta^2 \kappa L}{m^2} \mathbb{E}_{\boldsymbol{\xi}^{t-1}} \left\| \sum_{i=1}^{m} \tilde{\nabla}_{\boldsymbol{\theta}} g_i(\boldsymbol{\theta}_i^{t-1}, \boldsymbol{\lambda}_i^{t-1}) \right\|^2 \tag{77}$$

$$\leq \Phi(\bar{\boldsymbol{\theta}}^{t-1}) + \underbrace{\eta_\theta \langle \nabla \Phi(\bar{\boldsymbol{\theta}}^{t-1}), \nabla \Phi(\bar{\boldsymbol{\theta}}^{t-1}) - \frac{1}{m} \sum_{i=1}^{m} \nabla_{\boldsymbol{\theta}} g_i(\boldsymbol{\theta}_i^{t-1}, \boldsymbol{\lambda}_i^{t-1}) \rangle}_{:=T_4}$$

$$- \eta_\theta \nabla \| \Phi(\bar{\boldsymbol{\theta}}^{t-1}) \|^2 + \frac{\eta_\theta^2 \kappa L}{m^2} \underbrace{\mathbb{E}_{\boldsymbol{\xi}^t} \left\| \sum_{i=1}^{m} \tilde{\nabla}_{\boldsymbol{\theta}} g_i(\boldsymbol{\theta}_i^{t-1}, \boldsymbol{\lambda}_i^{t-1}) \right\|^2}_{:=T_5}. \tag{78}$$

We now turn bounding term $T_4$

$$T_4 = \eta_\theta \langle \nabla \Phi(\bar{\boldsymbol{\theta}}^{t-1}), \nabla \Phi(\bar{\boldsymbol{\theta}}^{t-1}) - \frac{1}{m} \sum_{i=1}^{m} \nabla_{\boldsymbol{\theta}} g_i(\boldsymbol{\theta}_i^{t-1}, \boldsymbol{\lambda}_i^{t-1}) \rangle \tag{79}$$

$$\stackrel{(19)}{\leq} \frac{\eta_\theta}{2} \left( \left\| \nabla \Phi(\bar{\boldsymbol{\theta}}^{t-1}) \right\|^2 + \left\| \nabla \Phi(\bar{\boldsymbol{\theta}}^{t-1}) - \frac{1}{m} \sum_{i=1}^{m} \nabla_{\boldsymbol{\theta}} g_i(\boldsymbol{\theta}_i^{t-1}, \boldsymbol{\lambda}_i^{t-1}) \right\|^2 \right) \tag{80}$$

$$\leq \frac{\eta_\theta}{2} \left( \left\| \nabla \Phi(\bar{\boldsymbol{\theta}}^{t-1}) \right\|^2 + \left\| \frac{1}{m} \sum_{i=1}^{m} \nabla_{\boldsymbol{\theta}} g_i(\bar{\boldsymbol{\theta}}^{t-1}, \boldsymbol{\lambda}^*(\bar{\boldsymbol{\theta}}^{t-1})) - \nabla_{\boldsymbol{\theta}} g_i(\boldsymbol{\theta}_i^{t-1}, \boldsymbol{\lambda}_i^{t-1}) \right\|^2 \right) \tag{81}$$

$$\overset{(12)}{\le} \frac{\eta_\theta}{2} \left( \left\| \nabla \Phi(\bar{\boldsymbol{\theta}}^{t-1}) \right\|^2 + \frac{L^2}{m} \sum_{i=1}^m \left\| \bar{\boldsymbol{\theta}}^{t-1} - \boldsymbol{\theta}_i^{t-1} \right\|^2 + \frac{L^2}{m} \sum_{i=1}^m \left\| \boldsymbol{\lambda}^*(\bar{\boldsymbol{\theta}}^{t-1}) - \boldsymbol{\lambda}_i^{t-1} \right\|^2 \right) \tag{82}$$

$$\overset{(20)}{\le} \frac{\eta_\theta}{2} \left( \left\| \nabla \Phi(\bar{\boldsymbol{\theta}}^{t-1}) \right\|^2 + \frac{L^2 \Xi_\theta^{t-1}}{m} + \frac{2L^2 \Xi_\lambda^{t-1}}{m} + \frac{2L^2}{m} \sum_{i=1}^m \underbrace{\left\| \boldsymbol{\lambda}^*(\bar{\boldsymbol{\theta}}^{t-1}) - \bar{\boldsymbol{\lambda}}^{t-1} \right\|^2}_{=\delta_\lambda^{t-1}} \right) \tag{83}$$

$$\tag{84}$$

and from stochastic gradient assumptions 3.4 we can bound $T_5$ as follows

$$T_5 = \mathbb{E}_{\boldsymbol{\xi}^{t-1}} \left\| \sum_{i=1}^m \tilde{\nabla}_{\boldsymbol{\theta}} g_i(\boldsymbol{\theta}_i^{t-1}, \boldsymbol{\lambda}_i^{t-1}) \right\|^2 \tag{85}$$

$$= \mathbb{E}_{\boldsymbol{\xi}^{t-1}} \left[ \left\| \sum_{i=1}^m \left( \tilde{\nabla}_{\boldsymbol{\theta}} g_i(\boldsymbol{\theta}_i^{t-1}, \boldsymbol{\lambda}_i^{t-1}) - \nabla_{\boldsymbol{\theta}} g_i(\boldsymbol{\theta}_i^{t-1}, \boldsymbol{\lambda}_i^{t-1}) \right) \right\|^2 \right] + \left\| \sum_{i=1}^m \nabla_{\boldsymbol{\theta}} g_i(\boldsymbol{\theta}_i^{t-1}, \boldsymbol{\lambda}_i^{t-1}) \right\|^2 \tag{86}$$

$$\overset{(20)}{\le} m\sigma_\theta^2 + 2 \left\| \sum_{i=1}^m \nabla_{\boldsymbol{\theta}} g_i(\boldsymbol{\theta}_i^{t-1}, \boldsymbol{\lambda}_i^{t-1}) - \nabla_{\boldsymbol{\theta}} g_i(\bar{\boldsymbol{\theta}}^{t-1}, \boldsymbol{\lambda}^*(\bar{\boldsymbol{\theta}}^{t-1})) \right\|^2 + 2 \left\| m \nabla \Phi(\bar{\boldsymbol{\theta}}^{t-1}) \right\|^2 \tag{87}$$

$$\overset{(12)}{\le} m\sigma_\theta^2 + 2L^2 m \sum_{i=1}^m \left\| \bar{\boldsymbol{\theta}}^{t-1} - \boldsymbol{\theta}_i^{t-1} \right\|^2 + 2L^2 m \sum_{i=1}^m \left\| \boldsymbol{\lambda}^*(\bar{\boldsymbol{\theta}}^{t-1}) - \boldsymbol{\lambda}_i^{t-1} \right\|^2 + 2m^2 \left\| \nabla \Phi(\bar{\boldsymbol{\theta}}^{t-1}) \right\|^2 \tag{88}$$

$$\le m\sigma_\theta^2 + 2L^2 m \Xi_\theta^{t-1} + 4L^2 m \Xi_\lambda^{t-1} + 4L^2 m \sum_{i=1}^m \underbrace{\left\| \boldsymbol{\lambda}^*(\bar{\boldsymbol{\theta}}^{t-1}) - \bar{\boldsymbol{\lambda}}^{t-1} \right\|^2}_{=\delta_\lambda^{t-1}} + 2m^2 \left\| \nabla \Phi(\bar{\boldsymbol{\theta}}^{t-1}) \right\|^2. \tag{89}$$

Recombining, grouping, and taking the expectation over the previous iterates we get the desired result. $\square$

**Lemma A.11.** *The sequence of $\{\delta_\lambda^t\}_{t=1}^T$ generated by Algorithm 1 satisfies*

$$\sum_{t=1}^T \mathbb{E}\left[\delta_\lambda^t\right] \le \frac{5\delta_\lambda^0 \kappa}{\eta_\lambda \mu} + \sum_{t=1}^T 5\kappa \left( 4D_\lambda^{t-1} \sqrt{\frac{1}{m} \mathbb{E}\left[\Xi_\theta^{t-1}\right]} + \frac{3\mathbb{E}\left[\Xi_\theta^{t-1}\right]}{m} + \frac{7\mathbb{E}\left[\Xi_\lambda^{t-1}\right]}{m} \right)$$
$$+ \sum_{t=1}^T 5 \left( \frac{8\kappa^2 \eta_\theta^2}{\eta_\lambda^2 \mu^2} \mathbb{E}\left[\left\| \nabla \Phi(\bar{\boldsymbol{\theta}}^{t-1}) \right\|^2\right] \right) + 5T \left( \frac{2\eta_\lambda \sigma_\lambda^2}{m\mu} + \frac{4\sigma_\theta^2}{16^2 m(\kappa+1)^2 \mu^2} \right) \tag{90}$$

*where $D_\lambda^{t-1} = \left\| \bar{\boldsymbol{\lambda}}^{t-1} - \boldsymbol{\lambda} \right\|$.*

**Proof:** From (20), for $b > 0$, we have

$$\mathbb{E}_{\boldsymbol{\xi}^{t-1}}[\delta_\lambda^t] \le \left(1 + \frac{1}{b}\right) \underbrace{\mathbb{E}_{\boldsymbol{\xi}^{t-1}} \left\| \boldsymbol{\lambda}^*(\bar{\boldsymbol{\theta}}^{t-1}) - \bar{\boldsymbol{\lambda}}^t \right\|^2}_{:=T_6} + (1+b) \underbrace{\mathbb{E}_{\boldsymbol{\xi}^{t-1}} \left\| \boldsymbol{\lambda}^*(\bar{\boldsymbol{\theta}}^t) - \boldsymbol{\lambda}^*(\bar{\boldsymbol{\theta}}^{t-1}) \right\|^2}_{:=T_7}. \tag{91}$$

Bounding $T_6$ similarly

$$T_6 = \mathbb{E}_{\boldsymbol{\xi}^{t-1}} \left\| \boldsymbol{\lambda}^*(\bar{\boldsymbol{\theta}}^{t-1}) - \bar{\boldsymbol{\lambda}}^t \right\|^2 \tag{92}$$

$$= \mathbb{E}_{\boldsymbol{\xi}^{t-1}} \left\| \boldsymbol{\lambda}^*(\bar{\boldsymbol{\theta}}^{t-1}) - \bar{\boldsymbol{\lambda}}^{t-1} - \frac{\eta_\lambda}{m} \sum_{i=1}^m \left( \tilde{\nabla}_{\boldsymbol{\lambda}} g_i(\boldsymbol{\theta}_i^{t-1}, \boldsymbol{\lambda}_i^{t-1}) \pm \nabla_{\boldsymbol{\lambda}} g_i(\boldsymbol{\theta}_i^{t-1}, \boldsymbol{\lambda}_i^{t-1}) \right) \right\|^2 \tag{93}$$

$$\le \left\| \boldsymbol{\lambda}^*(\bar{\boldsymbol{\theta}}^{t-1}) - \bar{\boldsymbol{\lambda}}^{t-1} - \frac{\eta_\lambda}{m} \sum_{i=1}^m \nabla_{\boldsymbol{\lambda}} g_i(\boldsymbol{\theta}_i^{t-1}, \boldsymbol{\lambda}_i^{t-1}) \right\|^2 + \frac{\eta_\lambda^2 \sigma_\lambda^2}{m} \tag{94}$$

$$= \left\| \boldsymbol{\lambda}^*(\bar{\boldsymbol{\theta}}^{t-1}) - \bar{\boldsymbol{\lambda}}^{t-1} \right\|^2 + \underbrace{\left\| \frac{\eta_\lambda}{m} \sum_{i=1}^m \nabla_{\boldsymbol{\lambda}} g_i(\boldsymbol{\theta}_i^{t-1}, \boldsymbol{\lambda}_i^{t-1}) \right\|^2}_{T_{6,1}}$$

$$\underbrace{-2\langle\boldsymbol{\lambda}^*(\bar{\boldsymbol{\theta}}^{t-1})-\bar{\boldsymbol{\lambda}}^{t-1};\frac{\eta_\lambda}{m}\sum_{i=1}^m\nabla_{\boldsymbol{\lambda}}g_i(\boldsymbol{\theta}_i^{t-1},\boldsymbol{\lambda}_i^{t-1})\rangle}_{T_{6,2}}+\frac{\eta_\lambda^2\sigma_\lambda^2}{m}. \tag{95}$$

Estimating $T_{6,1}$

$$T_{6,1}=2\eta_\lambda^2\left\|\frac{1}{m}\sum_{i=1}^m\nabla_{\boldsymbol{\lambda}}g_i(\boldsymbol{\theta}_i^{t-1},\boldsymbol{\lambda}_i^{t-1})\pm\nabla_{\boldsymbol{\lambda}}g_i(\bar{\boldsymbol{\theta}}^{t-1},\bar{\boldsymbol{\lambda}}^{t-1})-\nabla_{\boldsymbol{\lambda}}g_i(\bar{\boldsymbol{\theta}}^{t-1},\boldsymbol{\lambda}^*(\bar{\boldsymbol{\theta}}^{t-1}))\right\|^2 \tag{96}$$

$$\leq\frac{2\eta_\lambda^2}{m}\sum_{i=1}^m\left\|\nabla_{\boldsymbol{\lambda}}g_i(\boldsymbol{\theta}_i^{t-1},\boldsymbol{\lambda}_i^{t-1})-\nabla_{\boldsymbol{\lambda}}g_i(\bar{\boldsymbol{\theta}}^{t-1},\bar{\boldsymbol{\lambda}}^{t-1})\right\|^2$$

$$+2\eta_\lambda^2\left\|\frac{1}{m}\sum_{i=1}^m\nabla_{\boldsymbol{\lambda}}g_i(\bar{\boldsymbol{\theta}}^{t-1},\bar{\boldsymbol{\lambda}}^{t-1})-\nabla_{\boldsymbol{\lambda}}g_i(\bar{\boldsymbol{\theta}}^{t-1},\boldsymbol{\lambda}^*(\bar{\boldsymbol{\theta}}^{t-1}))\right\|^2 \tag{97}$$

$$\overset{(12,16)}{\leq}\frac{2\eta_\lambda^2}{m}\sum_{i=1}^m L^2\left\|\boldsymbol{\lambda}_i^{t-1}-\bar{\boldsymbol{\lambda}}^{t-1}\right\|^2+L^2\left\|\boldsymbol{\theta}_i^{t-1}-\bar{\boldsymbol{\theta}}^{t-1}\right\|^2+\frac{4\eta_\lambda^2 L}{m}\sum_{i=1}^m\left[g_i(\bar{\boldsymbol{\theta}}^{t-1},\boldsymbol{\lambda}^*(\bar{\boldsymbol{\theta}}^{t-1}))-g_i(\bar{\boldsymbol{\theta}}^{t-1},\bar{\boldsymbol{\lambda}}^{t-1})\right] \tag{98}$$

$$=\frac{2\eta_\lambda^2 L^2}{m}\Xi_\lambda^{t-1}+\frac{2\eta_\lambda^2 L^2}{m}\Xi_\theta^{t-1}+\frac{4\eta_\lambda^2 L}{m}\sum_{i=1}^m\left[g_i(\bar{\boldsymbol{\theta}}^{t-1},\boldsymbol{\lambda}^*(\boldsymbol{\theta}_i^{t-1}))-g_i(\bar{\boldsymbol{\theta}}^{t-1},\bar{\boldsymbol{\lambda}}^{t-1})\right]. \tag{99}$$

Estimating $T_{6,2}$

$$T_{6,2}=-2\frac{\eta_\lambda}{m}\sum_{i=1}^m\langle\boldsymbol{\lambda}^*(\bar{\boldsymbol{\theta}}^{t-1})-\bar{\boldsymbol{\lambda}}^{t-1};\nabla_{\boldsymbol{\lambda}}g_i(\boldsymbol{\theta}_i^{t-1},\boldsymbol{\lambda}_i^{t-1})\rangle \tag{100}$$

$$=-2\frac{\eta_\lambda}{m}\sum_{i=1}^m\langle\boldsymbol{\lambda}^*(\bar{\boldsymbol{\theta}}^{t-1})-\bar{\boldsymbol{\lambda}}^{t-1};\nabla_{\boldsymbol{\lambda}}g_i(\boldsymbol{\theta}_i^{t-1},\boldsymbol{\lambda}_i^{t-1})\pm\nabla_{\boldsymbol{\lambda}}g_i(\bar{\boldsymbol{\theta}}^{t-1},\boldsymbol{\lambda}_i^{t-1})\rangle \tag{101}$$

$$=-2\frac{\eta_\lambda}{m}\sum_{i=1}^m\langle\boldsymbol{\lambda}^*(\bar{\boldsymbol{\theta}}^{t-1})-\bar{\boldsymbol{\lambda}}^{t-1};\nabla_{\boldsymbol{\lambda}}g_i(\bar{\boldsymbol{\theta}}^{t-1},\boldsymbol{\lambda}_i^{t-1}\rangle$$

$$+2\frac{\eta_\lambda}{m}\sum_{i=1}^m\langle\bar{\boldsymbol{\lambda}}^{t-1}-\boldsymbol{\lambda}^*(\bar{\boldsymbol{\theta}}^{t-1});\nabla_{\boldsymbol{\lambda}}g_i(\boldsymbol{\theta}_i^{t-1},\boldsymbol{\lambda}_i^{t-1})-\nabla_{\boldsymbol{\lambda}}g_i(\bar{\boldsymbol{\theta}}^{t-1},\boldsymbol{\lambda}_i^{t-1})\rangle \tag{102}$$

$$\leq-2\frac{\eta_\lambda}{m}\sum_{i=1}^m\langle\boldsymbol{\lambda}^*(\bar{\boldsymbol{\theta}}^{t-1})-\bar{\boldsymbol{\lambda}}^{t-1};\nabla_{\boldsymbol{\lambda}}g_i(\bar{x}^{t-1},\boldsymbol{\lambda}_i^{t-1}\rangle+2\eta_\lambda LD_\lambda^{t-1}\sqrt{\frac{1}{m}\Xi_\theta^{t-1}} \tag{103}$$

$$=-2\frac{\eta_\lambda}{m}\sum_{i=1}^m\langle\boldsymbol{\lambda}^*(\bar{\boldsymbol{\theta}}^{t-1})-\boldsymbol{\lambda}_i^{t-1};\nabla_{\boldsymbol{\lambda}}g_i(\bar{\boldsymbol{\theta}}^{t-1},\boldsymbol{\lambda}_i^{t-1})\rangle+\langle\boldsymbol{\lambda}_i^{t-1}-\bar{\boldsymbol{\lambda}}^{t-1});\nabla_{\boldsymbol{\lambda}}g_i(\bar{\boldsymbol{\theta}}^{t-1},\boldsymbol{\lambda}_i^{t-1})\rangle$$

$$+2\eta_\lambda LD_\lambda^{t-1}\sqrt{\frac{1}{m}\Xi_\theta^{t-1}} \tag{104}$$

$$\overset{(15,18)}{\leq}2\frac{\eta_\lambda}{m}\sum_{i=1}^m\left(g_i(\bar{\boldsymbol{\theta}}^{t-1},\bar{\boldsymbol{\lambda}}^{t-1})-g_i(\boldsymbol{\theta}_i^{t-1},\boldsymbol{\lambda}^*(\bar{\boldsymbol{\theta}}^{t-1}))\right)+2\eta_\lambda LD_\lambda^{t-1}\sqrt{\frac{1}{m}\Xi_\theta^{t-1}}$$

$$-2\frac{\eta_\lambda}{m}\sum_{i=1}^m\left(\frac{\mu}{2}\left\|\boldsymbol{\lambda}^*(\bar{\boldsymbol{\theta}}^{t-1})-\boldsymbol{\lambda}_i^{t-1}\right\|^2+\frac{L}{2}\left\|\bar{\boldsymbol{\lambda}}^{t-1}-\boldsymbol{\lambda}_i^{t-1}\right\|^2\right) \tag{105}$$

$$\overset{(20)}{\leq}2\frac{\eta_\lambda}{m}\sum_{i=1}^m\left(g_i(\bar{\boldsymbol{\theta}}^{t-1},\bar{\boldsymbol{\lambda}}^{t-1})-g_i(\bar{\boldsymbol{\theta}}^{t-1},\boldsymbol{\lambda}^*(\bar{\boldsymbol{\theta}}^{t-1}))\right)+2\eta_\lambda LD_\lambda^{t-1}\sqrt{\frac{1}{m}\Xi_\theta^{t-1}}$$

$$-2\frac{\eta_\lambda}{m}\sum_{i=1}^m\left(\frac{\mu}{4}\left\|\boldsymbol{\lambda}^*(\bar{\boldsymbol{\theta}}^{t-1})-\bar{\boldsymbol{\lambda}}^{t-1}\right\|^2-\frac{L+\mu}{2}\left\|\bar{\boldsymbol{\lambda}}^{t-1}-\boldsymbol{\lambda}_i^{t-1}\right\|^2\right) \tag{106}$$

$$= -\frac{\mu\eta_\lambda}{2}\left\|\boldsymbol{\lambda}^*(\bar{\boldsymbol{\theta}}^{t-1}) - \bar{\boldsymbol{\lambda}}^{t-1}\right\|^2 - 2\frac{\eta_\lambda}{m}\sum_{i=1}^{m} g_i(\bar{\boldsymbol{\theta}}^{t-1}, \boldsymbol{\lambda}^*(\bar{\boldsymbol{\theta}}^{t-1})) - g_i(\bar{\boldsymbol{\theta}}^{t-1}, \bar{\boldsymbol{\lambda}}^{t-1})$$

$$+ \frac{2L\eta_\lambda}{m}\Xi_\lambda^{t-1} + 2\eta_\lambda LD_\lambda^{t-1}\sqrt{\frac{1}{m}\Xi_\theta^{t-1}} \tag{107}$$

where the last inequality follows from choosing $\eta_\lambda \leq 1/(2L)$. Substituting the expressions we get

$$T_6 = \left(1 - \frac{\mu\eta_\lambda}{2}\right)\left\|\boldsymbol{\lambda}^*(\bar{\boldsymbol{\theta}}^{t-1}) - \bar{\boldsymbol{\lambda}}^{t-1}\right\|^2 + \frac{\eta_\lambda^2\sigma_\lambda^2}{m} + \frac{L\eta_\lambda}{m}\Xi_\theta^{t-1} + \frac{3L\eta_\lambda}{m}\Xi_\lambda^{t-1} + 2\eta_\lambda LD_\lambda^{t-1}\sqrt{\frac{1}{m}\Xi_\theta^{t-1}}. \tag{108}$$

Being $\boldsymbol{\lambda}^*(\cdot)$ is $\kappa$-smooth (Lemma 4.3 (Lin et al., 2020a)) we can bound $T_7$ as follows

$$T_7 = \mathbb{E}_{\boldsymbol{\xi}^{t-1}}\left\|\boldsymbol{\lambda}^*(\bar{\boldsymbol{\theta}}^t) - \boldsymbol{\lambda}^*(\bar{\boldsymbol{\theta}}^{t-1})\right\|^2 \tag{109}$$

$$\leq \kappa^2\mathbb{E}_{\boldsymbol{\xi}^{t-1}}\left\|\bar{\boldsymbol{\theta}}^t - \bar{\boldsymbol{\theta}}^{t-1}\right\|^2 \tag{110}$$

$$= \frac{\kappa^2\eta_\theta^2}{m^2}\mathbb{E}_{\boldsymbol{\xi}^{t-1}}\left\|\sum_{i=1}^{m}\tilde{\nabla}_{\boldsymbol{\theta}}g_i(\boldsymbol{\theta}_i^{t-1}, \boldsymbol{\lambda}_i^{t-1})\right\|^2 \tag{111}$$

$$= \frac{\kappa^2\eta_\theta^2}{m^2}\left(m\sigma_\theta^2 + \left\|\sum_{i=1}^{m}\nabla_{\boldsymbol{\theta}}g_i(\boldsymbol{\theta}_i^{t-1}, \boldsymbol{\lambda}_i^{t-1}) \pm m\nabla\Phi(\bar{\boldsymbol{\theta}}^{t-1})\right\|^2\right) \tag{112}$$

$$\overset{(20)}{\leq} \frac{\kappa^2\eta_\theta^2}{m^2}\left(m\sigma_\theta^2 + 2\left\|m\nabla\Phi(\bar{\boldsymbol{\theta}}^{t-1})\right\|^2 + 2L^2m\sum_{i=1}^{m}\left\|\bar{\boldsymbol{\theta}}^{t-1} - \boldsymbol{\theta}_i^{t-1}\right\|^2 + 2L^2m\sum_{i=1}^{m}\left\|\boldsymbol{\lambda}^*(\bar{\boldsymbol{\theta}}^{t-1}) - \boldsymbol{\lambda}_i^{t-1}\right\|^2\right) \tag{113}$$

$$= \frac{\kappa^2\eta_\theta^2}{m^2}\left(m\sigma_\theta^2 + 2m^2\left\|\nabla\Phi(\bar{\boldsymbol{\theta}}^{t-1})\right\|^2 + 2L^2m\Xi_\theta^{t-1} + 2L^2m\sum_{i=1}^{m}\left\|\boldsymbol{\lambda}^*(\bar{\boldsymbol{\theta}}^{t-1}) - \bar{\boldsymbol{\lambda}}^{t-1} + \bar{\boldsymbol{\lambda}}^{t-1} - \boldsymbol{\lambda}_i^{t-1}\right\|^2\right) \tag{114}$$

$$\overset{(20)}{\leq} \kappa\eta_\theta^2\left(\frac{\sigma_\theta^2}{m} + 2\left\|\nabla\Phi(\bar{\boldsymbol{\theta}}^{t-1})\right\|^2 + \frac{2L^2\Xi_\theta^{t-1}}{m} + \frac{4L^2\Xi_\lambda^{t-1}}{m} + 4L^2\left\|\boldsymbol{\lambda}^*(\bar{\boldsymbol{\theta}}^{t-1}) - \bar{\boldsymbol{\lambda}}^{t-1}\right\|^2\right) \tag{115}$$

$$= \kappa^2\eta_\theta^2\left(\frac{\sigma_\theta^2}{m} + 2\left\|\nabla\Phi(\bar{\boldsymbol{\theta}}^{t-1})\right\|^2 + \frac{2L^2\Xi_\theta^{t-1}}{m} + \frac{4L^2\Xi_\lambda^{t-1}}{m} + 4L^2\delta_\lambda^{t-1}\right). \tag{116}$$

Recombining and grouping we get

$$\delta_\lambda^t \leq \left(\left(1 + \frac{1}{b}\right)\left(1 - \frac{\mu\eta_\lambda}{2}\right) + 4(1+b)\kappa^2\eta_\theta^2 L^2\right)\delta_\lambda^{t-1} + 2\left(1 + \frac{1}{b}\right)\eta_\lambda LD_\lambda^{t-1}\sqrt{\frac{1}{m}\Xi_\theta^{t-1}}$$

$$+ \left(\left(1 + \frac{1}{b}\right)L\eta_\lambda + 2(1+b)\kappa^2\eta_\theta^2 L^2\right)\frac{\Xi_\theta^{t-1}}{m} + \left(1 + \frac{1}{b}\right)\frac{\eta_\lambda^2\sigma_\lambda^2}{m} + (1+b)\kappa^2\eta_\theta^2\frac{\sigma_\theta^2}{m}$$

$$+ \left(\left(1 + \frac{1}{b}\right)3L\eta_\lambda + 4(1+b)\kappa^2\eta_\theta^2 L^2\right)\frac{\Xi_\lambda^{t-1}}{m} + 2\kappa^2\eta_\theta^2(1+b)\left\|\nabla\Phi(\bar{\boldsymbol{\theta}}^{t-1})\right\|^2. \tag{117}$$

Setting $b = 2\left(\frac{2}{\eta_\lambda\mu} - 1\right) > 0$ we get the following inequalities

$$\left(1 + \frac{1}{b}\right)\left(1 - \frac{\eta_\lambda\mu}{2}\right) \leq \left(1 - \frac{\eta_\lambda\mu}{4}\right) \tag{118}$$

$$(1 + b) \leq \frac{4}{\eta_\lambda\mu} \tag{119}$$

$$\left(1 + \frac{1}{b}\right) \leq 2 \tag{120}$$

$$\tag{121}$$

that allows to simplify (117) as follows

$$\delta_\lambda^t \leq \left(1 - \frac{\eta_\lambda\mu}{4} + \frac{16\kappa^2\eta_\theta^2 L^2}{\eta_\lambda\mu}\right)\delta_\lambda^{t-1} + 4\eta_\lambda LD_\lambda^{t-1}\sqrt{\frac{1}{m}\Xi_\theta^{t-1}} + \left(2L\eta_\lambda + \frac{8\kappa^2\eta_\theta^2 L^2}{\eta_\lambda\mu}\right)\frac{\Xi_\theta^{t-1}}{m}$$

$$+2\frac{\eta_\lambda^2\sigma_\lambda^2}{m} + \frac{4\kappa^2\eta_\theta^2\sigma_\theta^2}{m\eta_\lambda\mu} + \left(6L\eta_\lambda + \frac{16\kappa^2\eta_\theta^2 L^2}{\eta_\lambda\mu}\right)\frac{\Xi_\lambda^{t-1}}{m} + \frac{8\kappa^2\eta_\theta^2}{\eta_\lambda\mu}\left\|\nabla\Phi(\bar{x}^{t-1})\right\|^2. \tag{122}$$

Fixing $\eta_x = \frac{\eta_\lambda}{16(\kappa+1)^2}$ we get that

$$\nu = 1 - \frac{\eta_\lambda\mu}{4} + \frac{16\kappa^2\eta_x^2 L^2}{\eta_\lambda\mu} \le \left(1 - \frac{\eta_\lambda\mu}{5}\right). \tag{123}$$

Taking the expectation over the current iterate and applying recursively the inequality we obtain

$$\begin{aligned}
\mathbb{E}_{\boldsymbol{\xi}^{t-1}}[\delta_\lambda^t] \le{}& \nu^t\delta_\lambda^0 + \sum_{i=0}^{t-1}\nu^{t-1-i}\left(\frac{8\kappa^2\eta_\theta^2}{\eta_\lambda\mu}\mathbb{E}_{\boldsymbol{\xi}^{t-1}}[\|\nabla\Phi(\bar{\theta}^{t-1})\|^2] + 2\frac{\eta_\lambda^2\sigma_\lambda^2}{m} + \frac{4\kappa^2\eta_\theta^2}{\eta_\lambda\mu}\frac{\sigma_\theta^2}{m}\right)\\
&+ \sum_{i=0}^{t-1}\nu^{t-1-i}\left(4\eta_\lambda LD_\lambda^{t-1}\sqrt{\frac{1}{m}\mathbb{E}_{\boldsymbol{\xi}^{t-1}}[\Xi_\theta^{t-1}]}\right)\\
&+ \sum_{i=0}^{t-1}\nu^{t-1-i}\left(\frac{3L\eta_\lambda\mathbb{E}_{\boldsymbol{\xi}^{t-1}}[\Xi_\theta^{t-1}]}{m} + \frac{7L\eta_\lambda\mathbb{E}_{\boldsymbol{\xi}^{t-1}}[\Xi_\lambda^{t-1}]}{m}\right).
\end{aligned} \tag{124}$$

Summing from $t=1$ to $T$ and from (123) we get

$$\begin{aligned}
\sum_{t=1}^T\mathbb{E}_{\boldsymbol{\xi}^{t-1}}[\delta_\lambda^t] \le{}& \frac{5\delta_\lambda^0}{\eta_\lambda\mu} + \sum_{t=1}^T\frac{5}{\eta_\lambda\mu}\left(\frac{8\kappa^2\eta_\theta^2}{\eta_\lambda\mu}\mathbb{E}_{\boldsymbol{\xi}^{t-1}}[\|\nabla\Phi(\bar{\theta}^{t-1})\|^2]\right) + \frac{5T}{\eta_\lambda\mu}\left(2\frac{\eta_\lambda^2\sigma_\lambda^2}{m} + \frac{4\kappa^2\eta_\theta^2}{\eta_\lambda\mu}\frac{\sigma_\theta^2}{m}\right)\\
&+ \sum_{t=1}^T\frac{5}{\eta_\lambda\mu}\left(4\eta_\lambda LD_\lambda^{t-1}\sqrt{\frac{1}{m}\mathbb{E}_{\boldsymbol{\xi}^{t-1}}[\Xi_\theta^{t-1}]}\right)\\
&+ \sum_{t=1}^T\frac{5}{\eta_\lambda\mu}\left(\frac{3L\eta_\lambda\mathbb{E}_{\boldsymbol{\xi}^{t-1}}[\Xi_\theta^{t-1}]}{m} + \frac{7L\eta_\lambda\mathbb{E}_{\boldsymbol{\xi}^{t-1}}[\Xi_\lambda^{t-1}]}{m}\right).
\end{aligned} \tag{125}$$

$$\square$$

