# OpenReview forum: "Communication-Efficient Distributionally Robust Decentralized Learning"
_TMLR — Accepted by TMLR_

### Review · Reviewer_Btiw · 2022-10-18

**Summary Of Contributions:**

This paper proposes and analyzes a new method, AD-GDA, for decentralized distributionally robust learning. The method incorporates update compression (e.g., via quantization or sparsification) to reduce communication overhead. Theoretical convergence rates are derived for the non-convex setting, and experiments also illustrate the benefits of the proposed approach.

**Audience:**

Yes

**Broader Impact Concerns:**

None noted

**Claims And Evidence:**

Yes

**Requested Changes:**

1. It would be useful to more clearly explain what are the main challenges, if any, in deriving the theoretical results, or if any new analysis techniques are introduced in this work. (I realize this is not a requirement for publication in TMLR, just asking that the paper more clearly spell out if there is any contribution on this front.)
2. Consider better aligning the experiments with the theory by comparing empirical performance with that predicted by the theory for the different scenarios used in the paper.
3. Describe in greater detail how the baselines and AD-GDA were tuned in the experiments, both to enable a reader to reproduce your results and to reassure the reader that the comparison being made is fair.
4. Please expand the discussion of related work to include other papers on decentralized games and min-max optimization. The DRO problem is a particular instance of a min-max problem. For example, the paper [Stochastic Gradient Descent-Ascent and Consensus Optimization for Smooth Games: Convergence Analysis under Expected Co-coercivity](https://proceedings.neurips.cc/paper/2021/hash/9f96f36b7aae3b1ff847c26ac94c604e-Abstract.html) analyzes a similar method, but without communication compression. Is there particular problem structure that AD-GDA exploits to obtain new convergence guarantees, or is the main difference the inclusion of compression and error feedback?

**Strengths And Weaknesses:**

## Strengths
* Overall the paper is well-written.
* The theoretical rates match what one would expect for the non DRO setting order-wise, and they also match the upper bounds for a federated DRO method, Local SGDA, the client-server setting.
* Experiments illustrate the promise of the approach for applications.

## Weaknesses
* In terms of theory, the paper appears to make heavy use of existing well-known techniques, like prior analysis of SGDA and analyses incorporating error compensation in the context of compressed decentralized optimization.
* Although the experiments illustrate the usefulness of AD-GDA, they don't directly validate the theory, in the sense that there aren't direct comparisons to the way that the theory predicts rates should depend on key parameters like the level of compression or topology. Presently the comparison is mainly qualitative.
* The experiments section doesn't describe how different methods, including the baseline, were tuned.
* The literature review could be expanded to include additional related work on decentralized min-max optimization.

---

> ### Author Response · Authors · 2022-10-25
> **Response after comments**
>
> Thanks for the comments. We have updated the manuscript according to your suggestions, in particular:
>
> 1. We have updated the theory section and highlighted the challenges that are peculiar to the derivation of AD-GDA’s convergence guarantees together with the technical tools employed. While in the convex scenario the derivation is rather straightforward, in the non-convex case turns out that it is critical to bound the quantity
>  $\delta_\lambda^t:=\lVert{\lambda}^*(\bar{{\theta}}^{t})-\bar{{\lambda}}^{t}\lVert^2$.
> which represent the distance between ${\lambda}^*(\bar{{\theta}}^{t})$, the optimal dual variable for the *averaged* primal estimate, and $\bar{{\lambda}}^{t}$, the current *averaged* dual estimate. This key quantity is specific to the decentralized setting, and it is due to the existence of multiple estimates of  $\lambda$ and $\theta$ in the network. We have provided Lemma 4.2. to control $\delta_\lambda^t$, and we have highlighted the necessity of tuning the primal and dual learning rate to ensure that the inner optimization problem does not change too quickly and establish convergence.
>
> 2. We have replaced Figures 3 and 4 with convergence plots that highlight the predicted sub-linear rate of AD-GDA as well as the slope dependence on the compression levels and spectral gaps of the mixing matrices.
>
> 3. We have added more details about the experiments. We agree that experiment reproducibility is a fundamental aspect of research, and we would like to bring to your attention the supplementary material of the submission. This contains the implementation of AD-GDA (as well as DRFA and DR-DSGD) and all the necessary code/information to reproduce the large-scale experiments. We are planning to publish the code on GitHub too.
>
> 4. We have expanded the related work section and included the suggested work. The main difference between AD-GDA and *Stochastic Gradient Descent-Ascent and Consensus Optimization for Smooth Games: Convergence Analysis under Expected Co-coercivity* is that AD-GDA is a fully decentralized algorithm, meaning that there exist multiple primal and dual variables estimates across the network. This introduces the challenges and technicalities introduced in point 1.
>
> 5. To better state what are the differences/improvements between the proposed and existing decentralized DRO algorithms, we have added Table 1 and Section 4.1.
>
> Thanks again for the constructive comments, we hope that the above addresses your concerns. If not, we would be happy to follow up on further comments/request.

---

### Review · Reviewer_87oT · 2022-10-21

**Summary Of Contributions:**

**Summary**

The authors study the problem of distributionally-robust learning in the fully decentralized setting (i.e., in the absence of a network-wide coordinator), and propose a communication-efficient algorithm to solve this problem.

Solutions to this specific problem have been studied previously in the literature. The key difference in the authors' approach is an added emphasis on reducing the number of bits exchanged between nodes in the network. To achieve this, the authors use the Choco-SGD quantization technique introduced in Koloskova et al  (2019) for compressed communication between neighbors. Another difference between previous works in this area is the use of dual variable updates (as opposed to KL regularization) to address the minimax nature of the optimization problem.

The authors support their new algorithm (called agnostic decentralized gradient descent-ascent, or AD-GDA) with theoretical analysis under certain assumptions, and empirical demonstrations on FashionMNIST, CIFAR-10, and a microscopy image dataset.

**Audience:**

Yes

**Broader Impact Concerns:**

None.

**Claims And Evidence:**

No

**Requested Changes:**

**Critical**

* Consider rewriting the theory sections, drawing a sharper line between previously known results and new techniques/insights provided by the authors.
* It may be helpful to draw a sharper line between two closely related papers (DR-DSGD, DRFA) and how the proposed approach improves upon these works. In particular, DRFA also addresses distributional robustness as well as communication efficiency. Perhaps a table in the Introduction comparing the 3 methods in theory, and what gap is filled by AD-GDA that DRFA did not fill before, would be helpful.
* In Tables 1 and 2, the comparisons to Choco-SGD is fine but this is obviously not designed for robustness; it may be also helpful to report numbers for DR-DSGD and DRFA.

**Other comments**

* Is there any dependence on the number of workers ($m$) in Eq 11? Might be helpful to clarify.



**Strengths And Weaknesses:**

**Strengths**

* The authors study an interesting problem setup of importance to the community.
* The proposed technique is intuitive and easy to implement.

**Weaknesses**

* The theoretical contributions appear to be concatenations of existing techniques (analysis of decentralized SGD with dual variable updates, Choco-SGD). As such this is fine since the proofs seem correct, but they are presented without intuition. The authors could perhaps clarify what parts were known before, the technical difficulties in bringing them together, and any insights provided over and above existing results.
* It is unclear why theory results are provided for convex loss functions when none of the experiments involve convexity. (This might be a broader criticism applicable to papers in this area.)
* It is unclear why results are reported only on small-ish datasets like FashionMNIST and CIFAR-10. (This might be a broader criticism applicable to papers in this area.)

---

> ### Author Response · Authors · 2022-10-25
> **Revision after comments**
>
> Thanks for the comments. We have updated the manuscript according to your suggestions, in particular:
>
> 1. We modified the theory part, including a section in which we explained the difference between the existing distributionally robust algorithms and the proposed one. Furthermore, we highlighted the challenges that are peculiar to the derivation of AD-GDA’s convergence guarantees together with the technical tools employed. While in the convex scenario the derivation is rather straightforward, in the non-convex case turns out that it is critical to bound the quantity
>  $\delta_\lambda^t:=\lVert{\lambda}^*(\bar{{\theta}}^{t})-\bar{{\lambda}}^{t}\lVert^2$.
> which represent the distance between ${\lambda}^*(\bar{{\theta}}^{t})$, the optimal dual variable for the *averaged* primal estimate, and $\bar{{\lambda}}^{t}$, the current *averaged* dual estimate. This key quantity is specific to the decentralized setting, and it is due to the existence of multiple estimates of  $\lambda$ and $\theta$ in the network. We provide Lemma 4.2. to control $\delta_\lambda^t$, and we highlight the necessity of tuning the primal and dual learning rate to ensure that the inner optimization problem does not change too quickly and establish convergence.
>
> 2. We included Table 1 which compares the main features and convergence rates of the AD-GDA, DRFA and DR-DSGD. In summary, AD-GDA is the only decentralized and distributionally robust algorithm that can be used with any strongly concave regularizer and with message compression. Furthermore, AD-GDA enjoys a convergence rate that matches or outperforms the known ones.
>
> 3. Table 1 & 2 (Table 2 & 3 in the newer version) reports the worst-node accuracy of AD-GDA and CHOCO-SGD for different compression schemes and ratios. Since DRFA and DR-DSGD do not use compression, we believe that these tables are not the right place to benchmark these solutions. Instead, we decided to add a new Table in section 5.2.2 in which we reported the worst-node accuracy of AD-GDA, DR-DSGD and DRFA as requested. These results are given for the larger scaler experiments. Note that the F-MNIST setup is the same as the one in Table 1 & 2 (Table 2 & 3 in the newer version), just with more users. The bottom line of these additional experiments is that AD-GDA can efficiently find a distributionally robust solution and outperforms both baselines. We attribute these results to AD-GDA’s ability to solve the distributionally robust for any regularized (beyond the KL of DR-DSGD) without making use of the random sampling approach of DRFA.
>
> We have also clarified the independence of the convergence guarantee for the convex setting on the number of nodes. Regarding the weaknesses, we would like to highlight that solving the minimax problem in a decentralized manner introduces new challenges that we have detailed in the revised manuscript (see point 1 of this response). We would also like to point out that we have provided results for the logistic model which is a convex setting.
>
> Thanks again for the constructive comments, we hope that the above addresses your concerns. If not, feel free to reach out with further questions/requests.

---

### Review · Reviewer_2qTW · 2022-10-29

**Summary Of Contributions:**

## Summary of the paper
This paper studies the problem of distributionally robust optimization on graph networks with arbitrary connected topologies. Compared to other recent papers on the topic, it considers arbitrary regularizer, such as KL divergence, and covers both convex and non-convex functions. The authors propose an algorithm called AG-GDA that was derived as a method for the minmax problem arising in distributionally robust optimization. The theory is supported by experiments with different topologies, objectives and with different quantization.

## Summary of the review
The motivation for this work is a little bit unclear to me. Moreover, some of the assumptions seem to contradict specific choices of functions (KL divergence). On top of that, there do not seem to be novel ideas in this paper. The numerical evaluation, on the other hand, seems quite nice, even though it is mostly meant to support the theory.

**Audience:**

Yes

**Broader Impact Concerns:**

This work is mostly theoretical, so there are no broader impact concerns.

**Claims And Evidence:**

No

**Requested Changes:**

## Questions
1. What is exactly novel in terms of ideas in this paper?
2. Why the assumptions on bounded gradient do not contradict choosing $r$ as the KL divergence?
3. What would be an interesting example of $r$ that is not KL?

## Writing issues
Some notation feels unnecesseary, for instance, it is introduced for some reason that $f_i$ is given by $\mathbb{E}_{\mathbb{R}z\sim P_i}\ell(\theta, \mathbb{R}z)$, but as far as I can see, $\ell$ is not used anywhere
Assumption 3.1 should include the definition of eigengap, I do not see where it is explained
I think in Assumption 3.2, delta should from the range $(0, 1]$ rather than $[0, 1]$
"gradient based algorithm" ---> "gradient-based algorithm"
In Algorithm 1, the Gossip step uses $s_i^{t+1}$, which is defined only a few lines later, I think it should be $s_i^t$
The projection step of Algorithm 1 is not explained anywhere
The text in Theorem 4.3 states that the output point $\theta_o$ satisfies a certain property, but the mathematical bound is for the average iterates $\overline \theta^{t-1}$
Page 11, "CHOCOSGD represent" ---> "CHOCOSGD represents"
In table, please highlight the best number in every column
It is confusing that Lemma A.6 and A.7 refer to the works of Koloskova et al. Their papers do not consider the iterates of Algorithm 1, for which the statements are claimed in Lemma A.6 and A.7. So please make these Lemams more specific and explain which results of Koloskova et al. you refer to
Some equations, for instance (27), (31), and (40), lack punctuation

**Strengths And Weaknesses:**

## Strengths
1. The considered problem might be of interest to the community working on distributed optimization as it formulates a clear problem and suggests a solution.
2. The theory covers both convex and nonconvex problems and the claimed rates look reasonable.
3. The experiments are done with multiple seeds and confidence intervals are reported. They are quite comprehensive and seem to support the theory.


## Weaknesses
1. One of the paper's contributions is to extend the decentralized distributionally robust optimization beyond Kullback-Leibler to arbitrary regularizers $r$, but it is not explained why we need any other regularizer. As far as I know, it is standard to consider KL regularization.
2. Assumption 3.4 states that the gradients remain bounded, but as far as I can see, this is not possible with KL regularization. In particular, when of $\lambda_i$ approaches 0, the gradient would explode.
3. Overall, the paper feels like a combination of two existing techniques, without strong practical motivation. It is not very clear where exactly is the novelty, all ideas seem to come from prior work.
4. The writing suffers from a lot of small issues.

---

> ### Author Response · Authors · 2022-10-31
> **Revision after comments**
>
> 1- *One of the paper's contributions is to extend the decentralized distributionally robust optimization beyond Kullback-Leibler to arbitrary regularizers r, but it is not explained why we need any other regularizer. As far as I know, it is standard to consider KL regularization. What would be an interesting example of r that is not KL?*
>
> The original distributionally robust objective, which results from bounding the excess risk of the distributionally robust learner, is based on chi-squared regularization (see Theorem 2 in the https://arxiv.org/abs/1902.00146 ) while KL regularization has been considered in DR-DSGD to sidestep the decentralized minimax problem.
> In this work, we *directly* tackle the decentralized minimax problem and generalize to any strongly concave regularizer, therefore including the chi-squared one. As a result, AD-GDA combined with the chi-squared regularized directly minimizes the distributional robust excess risk bound and it constitutes the first decentralized algorithm that exactly solves the distributionally robust problem. This translates into better performance compared to alternatives based on surrogate optimization problems, see Table 5 in the updated versions. Other than the chi-squared divergence, another interesting divergence that can serve as a regularizer is the Tsallis divergence. The Tsallis divergence is based on generalized logarithms (https://arxiv.org/pdf/1705.07210.pdf)  and it exhibits increased robustness to outliers. In this specific case could be used to produce procedures that are resilient to malicious or outlying nodes.
>
> 2- *Overall, the paper feels like a combination of two existing techniques, without strong practical motivation. It is not very clear where exactly is the novelty, all ideas seem to come from prior work. What is exactly novel in terms of ideas in this paper?*
>
> The main practical motivation is to perform distributionally robust learning in a decentralized manner. Robust and fair machine learning models are of utmost importance, especially in decentralized settings in which local distributions are often affected by local variables. Figure 2 constitutes a motivating example based on biological data.
> The main novelty of the paper consists in proposing the first decentralized algorithm to perform exactly distributionally robust learning, extending to all strongly concave regularizers, topologies beyond the star one and compression. Furthermore, AD-GDA enjoys a convergence rate that matches or outperforms the known ones.  In the revised manuscript. we have included Table 1 and Section 4.1 to make these points clearer.
> From a technical perspective, even if the algorithm looks simple, it comes with the difficulties of solving a minimax optimization problem in a decentralized manner (note that this was the main reason why DR-DSGD has been limited to KL regularizer). In the updated version we have updated the theory section and highlighted the challenges that are peculiar to the derivation of AD-GDA’s convergence guarantees together with the technical tools employed. While in the convex scenario the derivation is rather straightforward, in the non-convex case turns out that it is critical to bound the quantity
> $\delta_\lambda^t:=\lVert{\lambda}^*(\bar{{\theta}}^{t})-\bar{{\lambda}}^{t}\lVert^2$
> which represent the distance between ${\lambda}^*(\bar{{\theta}}^{t})$, the optimal dual variable for the *averaged* primal estimate, and $\bar{{\lambda}}^{t}$, the current *averaged* dual estimate. This key quantity is specific to the decentralized setting, and it is due to the existence of multiple estimates of  $\lambda$ and $\theta$ in the network. We have provided Lemma 4.2. to control $\delta_\lambda^t$, and we have highlighted the necessity of tuning the primal and dual learning rate to ensure that the inner optimization problem does not change too quickly and establish convergence.
>
> 3- *Why the assumptions on bounded gradient do not contradict choosing r as the KL divergence? Assumption 3.4 states that the gradients remain bounded, but as far as I can see, this is not possible with KL regularization. In particular, when of $λ_i$ approaches 0, the gradient would explode.*
>
> Yes, the KL divergence does not have a bounded gradient when $\lambda_i\to0$ and in practice a bound on the gradient norm could be obtained restricting the domain of lambda to $\lambda_i>\epsilon$. However, we would like to stress that this work extends beyond KL divergence and in our manuscript, we focus on divergences that indeed satisfy this constraint, e.g. the chi-squared divergence that always has a bounded gradient. Also, the Tsallis divergence, which we have envisioned before to improve outliers robustness, has a bounded gradient for $t>1$.
> Thanks again for the constructive comments and carefully reading the paper, we have fixed typos and we hope that the above addresses your concerns. If not, feel free to reach out with further questions/requests.

---

> > ### Comment · Reviewer_2qTW · 2022-11-03
> > **Thanks for the clarifications**
> >
> > 1. Your clarifications are very welcome. I can see now why having a unified perspective can be helpful.
> > 2. Thank you for explaining the technical difficulty.
> > 3. If KL is not supported due to divergence of gradients, I think the claim that the method "can be used with any strongly concave regularize" is an overstatement. Please make clear the limitations of the theory in the paper.

---

> > > ### Author Response · Authors · 2022-11-04
> > > **Revision**
> > >
> > > Thanks for the prompt reply, we are glad to have clarified your doubts.
> > > Regarding the limitations, we have uploaded a revised version and modified the manuscript as follows:
> > >
> > > 1. "AD-GDA directly tackles the distributionally robust problem and therefore can be applied to any strongly concave regularizer *that satisfies Assumption 3.4* "
> > > 2. "The bounded magnitude assumption is rather strong and *limits the choice of regularization functions*, but it is often made in distributed stochastic optimization"

---

### Decision · Action_Editors · 2022-12-24

**Recommendation:** Accept as is

**Comment:**

This paper proposes and analyzes a new method, AD-GDA, for decentralized distributionally robust learning. The main contribution over existing work is to simultaneously enable the fully decentralized setting along with general regularizers beyong KL divergence. The reviewers generally found the paper well written and the claims made in the paper well-supported by the evidence provided in the paper. Some reviewers had some concerns regarding the assumptions and the motivation for extending the previous work beyond KL which were addressed by the authors in the revisions. Overall all the reviewers recommended an accept or a leaning accept with novelty being the only concern area.

**Audience:**

Yes researchers working in optimization in a distributed setting will be interested in the paper.

**Claims And Evidence:**

The paper's primary contribution is an algorithm for distributed optimization. The paper provides convergence rate bounds which have been proven in the paper and the reviewers have verified its correctness. The paper provides some experimental evidence which supports their claims in the settings considered by the authors.